# Antimicrobial Nanotubes: From Synthesis and Promising Antimicrobial Upshots to Unanticipated Toxicities, Strategies to Limit Them, and Regulatory Issues

**DOI:** 10.3390/nano15080633

**Published:** 2025-04-21

**Authors:** Silvana Alfei, Gian Carlo Schito

**Affiliations:** 1Department of Pharmacy (DIFAR), University of Genoa, Viale Cembrano, 4, 16148 Genoa, Italy; 2Department of Surgical Sciences and Integrated Diagnostics (DISC), University of Genoa, Viale Benedetto XV, 6, 16132 Genova, Italy; giancarlo.schito@unige.it

**Keywords:** carbon nanotubes (CNTs), mechanical, electrical, and optical properties, CNTs biomedical applications, antibacterial nanotubes, nanotubes toxicity, CNT regulation

## Abstract

Nanotubes (NTs) are nanosized tube-like structured materials made from various substances such as carbon, boron, or silicon. Carbon nanomaterials (CNMs), including carbon nanotubes (CNTs), graphene/graphene oxide (G/GO), and fullerenes, have good interatomic interactions and possess special characteristics, exploitable in several applications because of the presence of sp2 and sp3 bonds. Among NTs, CNTs are the most studied compounds due to their nonpareil electrical, mechanical, optical, and biomedical properties. Moreover, single-walled carbon nanotubes (SWNTs) have, in particular, demonstrated high ability as drug delivery systems and in transporting a wide range of chemicals across membranes and into living cells. Therefore, SWNTs, more than other NT structures, have generated interest in medicinal applications, such as target delivery, improved imaging, tissue regeneration, medication, and gene delivery, which provide nanosized devices with higher efficacy and fewer side effects. SWNTs and multi-walled CNTs (MWCNTs) have recently gained a great deal of attention for their antibacterial effects. Unfortunately, numerous recent studies have revealed unanticipated toxicities caused by CNTs. However, contradictory opinions exist regarding these findings. Moreover, the problem of controlling CNT-based products has become particularly evident, especially in relation to their large-scale production and the nanosized forms of the carbon that constitute them. Important directive rules have been approved over the years, but further research and regulatory measures should be introduced for a safer production and utilization of CNTs. Against this background, and after an overview of CNMs and CNTs, the antimicrobial properties of pristine and modified SWNTs and MWCNTs as well as the most relevant in vitro and in vivo studies on their possible toxicity, have been reported. Strategies and preventive behaviour to limit CNT risks have been provided. Finally, a debate on regulatory issues has also been included.

## 1. Introduction

Nanotechnology is one of the most exciting 21st century technologies [1,2], with the capability to observe, measure, control, assemble, and manufacture materials at the nanoscale, and it translates the theory of nanoscience into practical applications [3,4,5]. Nanotechnology is a discipline of global interest, which is constantly growing [6,7,8]. It deals with a variety of materials produced at <100 nm through different chemical and physical methods [9]. It is conducted at the nanoscale (1–100 nm) and has unique phenomena that enable novel applications in a wide range of fields, including chemistry, physics, biology, medicine, engineering, and electronics [10,11]. To keep up with the rate of advancement in science and technology, new types of nanomaterials with unconventional edge and innovative properties are incessantly required [12]. Nanotubes (NTs) belong to a promising group of nanomaterials that allow us to approach several new electronic, magnetic, optical, and mechanical properties [13].

Many of the nanotube structures that have been extensively studied contain boron, silicon, and molybdenum atoms; carbon nanotubes (CNTs) are the most researched group among existing NT structures [14]. In fact, carbon nanomaterials (CNMs), including CNTs, graphene/graphene oxide (G/GO), and fullerenes, are allotropes of sp2 and sp3 hybridized carbon atoms and their sp2 and sp3 bonds and are thought to be responsible for their unique characteristics [12]. They have good interatomic interactions [14] and are very interesting nanostructures in terms of their special properties and possible applications [15]. CNTs are composed of graphite sheets rolled up in an unbreakable and non-stop hexagonal-like lattice structure, in which carbon atoms appear at the tops of the hexagon-type forms. Based on the number of carbon sheets, CNTs are categorized as single-wall carbon nanotubes (SWCNTs), double-wall carbon nanotubes (DWCNTs), and multi-wall carbon nanotubes (MWCNTs) [13]. Although MWCNTs have also been extensively studied for their demonstrated antimicrobial potential, SWCNTs have displayed excellent antibacterial action, according to some research, due to the smaller size of these materials, which play a significant part in the inhibition of microbes [12]. Through a search made on Scopus on 1 March 2025, using “nanotubes” as a keyword, for the last fifty years (2010–2025), we found 87,798 documents concerning “nanotubes” and 80,535 documents concerning “carbon nanotubes” (Figure 1). Through an additional survey and using the keywords “single-walled nanotubes”, we found 17,225 publications (Figure 1).

Generally, CNMs offer great surface area, mechanical resistance, thermal conductivity, photoluminescence, transparency, and constructional durability, in addition to antibacterial activities against pathogens and remarkable electrical conductivity [16]. These properties increasingly encourage the application of CNMs in several nanocomposites, such as in thin-film transistors, transparent conducting electrodes, photovoltaics, supercapacitors, biosensors, drug delivery systems, tissue engineering, photothermal therapy, and antimicrobial food packaging. As a part of the larger family of CNMs, CNTs are also materials with extraordinary properties that are useful across a variety of new, state-of-the-art applications in sensors, printed electronics, e-readers, flexible displays, energy storage medical treatments, and more. Specifically, since their discovery in 1991 by Ijima [17], SWNTs have influenced a significant amount of activity in both research and industry across the world. Furthermore, SWNTs have stimulated considerable investment in manufacturing methods, characterization techniques, and application development. However, the main drawback of CNTs is immediately recognized; it consists of their scarce solubility in most solvents, which limits their application [16]. To address this issue and amplify CNT use, researchers have focused on surface modifications [14,15,16]. Surface modifications, together with several other factors, including their chemical composition, target bacteria, and reaction environment, can also affect the CNMs’ antimicrobial activity [18]. The antimicrobial effects of CNMs are derived mainly from their capacity to physically isolate pathogen cells from their supportive environment. Anyway, their capacity to penetrate microbial cell walls/membranes, thus causing irreversible structural damage, supports their antimicrobial activity to a large extent [12]. Additionally, the interaction of CNMs with bacterial cells stimulates the generation of noxious substances, including reactive oxygen species (ROS), triggering oxidative stress (OS) in cells and thus promoting their death. Additionally, the interactions between CNMs and microorganisms result in an electron transfer, which induces ROS-independent OS and causes the biological death of pathogens [19]. To confirm that the OS plays an additional role in CNT and SWCNT antimicrobial mechanisms [20], Haung et al. investigated the mechanical effects that influenced the antimicrobial properties of CNTs, including low wear rates, low friction coefficients, favourable tribological characteristics, and high corrosion resistance [21]. Among different types of CNMs and CNTs, the antimicrobial activity of SWCNTs is found to be higher due to their advantageous physicochemical properties [22,23,24,25]. In this regard, Kang and co-workers, in their first report on the antimicrobial activity of purified SWCNTs and MWCNTs, demonstrated that both materials showed significant antimicrobial effects. Moreover, those of SWCNTs seemed to be stronger than those of MWCNTs [26]. They found that CNTs’ antibacterial effect mainly depended on their ruinous impact on the integrity of bacterial membranes upon direct contact; the higher activity of SWCNTs could be due to their small size, which provides a larger surface area to facilitate the membrane perturbation [26,27]. The morphology and metabolic activities of pathogens were also compromised [27]. Also, based on studies conducted by Chen et al., the SWCNTs played a significant role as “nano-darts”, which penetrated bacterial cell walls, reduced membrane potential, released intracellular constituents (DNA and RNA), and ultimately disrupted bacterial membranes [28]. Unfortunately, despite their possible toxicity being initially disregarded due to their harmless carbon-based composition, early and recent investigations have unexpectedly unveiled that CNTs could be toxic to humans, animals, and the environment, depending on their concentrations and exposure times [29]. In fact, even if CNTs are made of normally innocuous and intrinsically safe carbon and do not have nano-dimensioned length, they should be considered as nanomaterials, which, thanks to their nanosized diameter, can penetrate various cells, including those in bacteria, yeast, and mammals [30]. It has been found that, mainly due to their poor biodegradability, accumulation, and persistence in the body, surface structure, and residual impurities, CNTs can be remarkably toxic, genotoxic, and cancerogenic to several cells, organs, and animal models. In this regard, amyloid deposits have been found in the brain, liver, lungs, and kidneys of exposed animals [29,31]. Additionally, investigations prompted by the similarity of the needle-like CNT fibres with those of asbestos evidenced that they can be responsible for pulmonary toxicity, mainly in CNT manufacturers, which results from contact with and/or inhalation of CNT fibres [32,33]. Also, toxicity to the cardiovascular system, immune and reproductive systems, embryos, and neurotoxicity have been reported [33]. Based on these findings, several strategies using advanced synthetic methods, highly efficient purification procedures, post-synthesis chemical modifications, as well as preventive behavioural conducts have already been developed to limit CNTs’ toxic effects [34]. Anyway, more and incessant studies should be implemented to make their production and employment safer. On the other hand, despite significant advancements in terms of regulations concerning CNT production and possible applications for health and safety at the production sites being implemented, several unsolved regulatory issues survive, which need urgent solutions [34]. The increasing production and commercialization of differently structured CNTs and related nanocomposites and their daily application have stimulated the development of more rigid regulatory standards concerning their manufacturing and utilization [34]. Anyway, substantial concern regarding toxicity and environmental safety persists, mainly due to contradictory outcomes of toxicity investigations [35]. However, it must be recognized that CNTs, despite their still not fully clear risk to living organisms and the environment and limited regulation, are paradoxically ubiquitously exploitable for improving the quality of their life. In this regard, with this review, we aimed at stimulating further research on these nonpareil materials to better understand their potential as antimicrobial agents and the mechanisms supporting their cytotoxic effects on pathogens, which could also help to clarify the basis of their toxicity to humans. Such improved knowledge could allow their safer, large-scale production and utilization to enhance environment equilibrium and our quality of life. Against this background, and after an overview of CNMs and CNTs, the antimicrobial properties of pristine and modified SWCNTs and MWCNTs, as well as the most relevant in vitro and in vivo studies on their possible toxicity, have been discussed. The strategies already developed and the suggested preventive behaviour to limit CNT toxicity have also been provided and discussed. Finally, an extensive debate on the regulatory issues and the currently available standard guidelines by relevant International Organizations to produce, characterize, and manage CNTs and related nanocomposites have also been included.

This work is a thorough and contemporary review, which thoroughly explores and discusses different aspects concerning CNTs in a single document, including the synthesis, structure, applications, antimicrobial properties, functionalization strategies, toxicological concerns, and regulatory issues. This review affords systematic and well-organized information on CNTs and their potential applications. This study is supported by a substantial body of literature research and several case-specific tables, which enhance its readability and clarity, showing the double-sided impact that CNT use could have on our lives and providing future perspectives for the most challenging aspects of CNTs.

## 2. Approaching Carbon Nanotubes

Carbon is a chemical element that has atomic number 6. It is one of the most abundant elements in the Universe by mass, capable of providing approximately 10 million different pure organic compounds. Such compounds possess the uncommon ability to form polymers at earth temperatures, thus being the chemical basis of all known life [35].

CNTs are thin and long cylinders made of carbon, which were found for the first time in 1991 by Sumio Iijima [17]. Many physical properties of CNTs are still obscure and need to be disputed, as well as technical and not technical hurdles, which still limit the CNTs’ success and their extensive application. Anyway, it is generally recognized that CNTs possess a plethora of thermal, electronic, mechanical, and structural properties that can vary based on their different existing forms [35]. In this regard, CNTs differ mainly in their diameter, length, chirality, or twist. Table A1, in Appendix A and reproduced from our previous work [35], collects the key properties and potential uses of CNTs, as well as the current remaining technical and non-technical hurdles. References in Table A1 do not belong to this work but are those contained in Alfei and Schito, 2022 [35]. We have decided on this solution so as not to overly burden the already significantly long list of references in this work. In addition to those reported in Table A1, a plethora of other possible applications for CNTs exist, including solar collectors, conductive and/or waterproof paper, catalyst supports, nano-porous filters, and coatings of all types. Since CNTs possess the capability to absorb infrared light, they may be applied to the I/R Optics Industry. We are confident that many unexpected applications for these nonpareil materials will be found in the next years, which will reinforce the belief that CNTs could be the most important and valuable nanomaterial ever used until now.

### 2.1. Synthesis of CNTs

Table A2 in Appendix A, reproduced by our previous work [35] collects the descriptions of the three most used methods to synthetize CNTs. Briefly, they include arc discharge (AD) method [36,37], finalized to the production of fullerenes, before 1991 [17], laser ablation (LA) [38,39], developed by Dr. Richard Smalley and co-workers at Rice University, and chemical vapor deposition (CVD) [40,41,42,43,44,45,46], which is the most widely used method to produce CNTs [35]. An advanced CVD technique is named plasma-enhanced chemical vapor deposition (PECVD) and consists of generating plasma by the application of a strong electric field during CNTs growth [47]. By adapting the reactor’s geometry, it is possible to synthesize vertically aligned CNTs (VACNTs) [35], whose morphology is of interest to researchers attracted by electron emission from CNTs. Other methods have been described in detail in our previous work [35]. They include high-pressure carbon monoxide (HiPco) process, optimized at Rice University [35], super-growth CVD (SGCVD) [48,49], introduced by Kenji Hata, Sumio Iijima and col-leagues at AIST (Japan), plasma torch (PT) [50,51], an invention by Olivier Smiljanic in the year 2000, working at Institut National de la Recherche Scientifique (INRS) (Varennes, Canada), subsequently modified in IPT procedure by researchers from Sherbrooke University and the National Research Council of Canada and liquid electrolysis method (LEM) [52,53], which allowed to obtain MWCNTs by electrolysis of molten carbonates [35]. In addition to all these artificial inventions by human re-searcher, CNTs can form naturally in commonly originated flames emitted by burning methane [54], ethylene [55], and benzene [56]. Highly irregular in dimensions and low-quality CNT-based structures can form in smoke from both indoor and outdoor air [57]. Anyway, lacking the high degree of uniformity, due to uncontrolled conditions, necessary to satisfy the many needs of both research and industry, their practical ap-plication is hampered. Despite this, efforts focus on the strategies to control environ-mental flames by theoretical models to produce less irregular CNTs [58,59,60,61,62], thus going towards large-scale, low-cost CNTs synthesis competitive with large scale CVD pro-duction.

### 2.2. Architecture and Other Features of CNTs

CNTs deserved the prefix “nano” due to their very small size of 1/50,000th of the diameter of a human hair.

Pure carbon can develop in several carbon-based architectures, and CNTs are one of them. For example, diamond is the tetrahedron crystalline structure of carbon, while graphite or graphene owns a planar structure, with linked carbon atoms forming hexagons [35]. Buckyballs were named after F. Buckminister Fuller, who first designed geodesic spheres; they are also called fullerenes [63]. They are made of hexagons of pure carbon atoms linked to form a sphere. The typical structure of CNTs instead derives from the rolling up of a sheet of carbon atoms associated to form hexagons (graphene), thus forming cylinders (Figure 2a).

Different structures that CNTs can assume are represented in Figure 2b. While SWCNTs (left side) are not naturally formed CNTs, MWCNTs are more complex CNTs formed naturally [63].

Additionally, when CNTs consist of two or three graphene complete cylinders with one inside the other, they are termed double- and triple-walled carbon nanotubes (DWCNTs and TWCNTs). Figure 3 shows the representative structure of a DWCNT.

CNTs are, atomically speaking, very stable, while SWCNTs are more stable than MWCNTs. Researchers in the field consider tubes with diameters less than 100 nm as CNTs [64]. However, no intrinsic limit exists on how long CNTs can develop [35]. However, their length is usually much larger than their diameter [35]. Table A3 in Appendix A summarizes the physical limits of CNTs. The references in the table do not belong to this work but are those contained in Alfei and Schito, 2022 [35]. We have decided on this solution for the reasons previously described. The production of SWCNTs is more difficult than that of MWCNTs, which can exhibit behaviours different from SWCNTs. In this regard, despite the dispersion of SWCNTs to obtain nanocomposites being more difficult than that of MWCNTs, generally, SWCNTs demonstrated better properties than MWCNTs. For this reason, scientists are focused mainly in finding more practical ways to mass-produce SWCNTs. Concerning the production of CNT-based nanocomposites, the possible junctions between two or more CNTs have been widely discussed theoretically [65,66], while those between CNTs and graphene have been considered both theoretically [67] and experimentally [68] (Table A4, Appendix A, references in the Table do not belong to this work but are those contained in Alfei and Schito, 2022 [35], according to reasons previously described). Junctions between CNTs were observed in CNTs prepared by AD or CVD methods. Lambin et al. were the first experts who studied theoretically the electronic properties of such junctions, starting from the assumption that a connection between a metallic tube and a semiconducting one could form a component of a CNT-based electronic circuit [69]. CNTs-graphene junctions are instead the basis of pillared graphene, characterized by three-dimensional (3D)-carbon nanotube (3D-CNTs) architectures, studied as building blocks to fabricate three-dimensional macroscopic structures [70]. Using these materials, it is possible to obtain free-standing, porous scaffolds made only of carbon, possessing macro-, micro-, and nano-structured pores and tailorable porosity with several applications. They could be used for the fabrication of the next-generation energy storages, supercapacitors, field emission transistors, high-performance catalysts, photovoltaics, and biomedical devices, such as implants and biosensors [71,72,73].

#### Relationships Structure/Properties

The study of CNT formations remains a cutting-edge field where new discoveries are being made regularly. Among the possible intimate structures that a SWCNT can assume, the armchair, zigzag, and chiral configurations are extensively recognized, based on the graphene mode to package into the CNTs cylinder (Figure 4).

The structural conformation of CNTs has a direct effect on their properties, mainly on the mechanical and electrical ones. For example, in MWCNTs, the outer walls of MWCNTs can protect the inner carbon tubes from chemical interactions with outside materials. Over the years, possible structure-related mechanical properties and structure-related electrical and thermal properties of CNTs have been reported in several studies. Structure-related mechanical investigations established that CNTs are the greatest nanomaterials discovered so far. Particularly, CNTs were better than other nanomaterials in terms of elastic modulus and tensile strength [35,74,75,76]. Anyway, it has also been reported that defects in the structure of CNTs due to atomic vacancies or rearrangements of the carbon bonds can cause weak points in small segments of the CNT, which reduce their elasticity and weaken their tensile strength [35]. On the other hand, structure-related electrical and thermal investigations showed that the structure of CNTs affect their conductive potency, both in terms of electrical and thermal conductivity [35,77,78,79,80,81,82].

## 3. Biomedical Applications of CNTs

CNTs belonging to all categories, including SWCNTs, DWCNTs, and MWCNTs, can differ in purity, length, and functionality. Anyway, all own a plethora of properties, such as high electrical conductivity, high tensile strength, light weight [83,84,85], and high biocompatibility [86]. All possess the capability to load molecules for their transport and delivery, large surface area, and chemical inertness. All can be enriched with functional groups and demonstrated good elasticity, thermal conductivity, capability to expand, electron emission capacity, and high aspect ratio [83,84,85].

These properties, added to peculiar composition and geometry, make CNTs suitable for numerous potential applications, including energy storage, biomedical uses, and air and water filtration. Also, CNTs could become molecular electronics, thermal materials, structural materials, electrical conductors, fabrics and fibres, catalyst supports, conductive plastics, conductive adhesives, as well as ceramics [83].

For this reason, scientists working in this sector are involved in efforts aimed at increasingly lowering the costs of CNT production to commercially viable levels by scaling up their synthesis.

Interestingly, CNTs have great potential to be applied in nanomedicine for disease diagnosis and drug targeting, as well as to transport various biomolecules such as proteins, DNA, RNA, immune-active compounds, and lectins [87,88,89].

CNTs are also extensively applied to develop electrochemical sensors, DNA-based sensors, as well as piezoelectric and gas sensors [83]. Additionally, opportunely structured CNTs, such as cationic CNTs, have been revealed to possess interesting antibacterial and antifungal activity [83]. Table A5 in Appendix A, reproduced from our previous article [35] summarizes the several applications that CNTs could have in the biomedical area. References in the Table do not belong to this work but are those contained in Alfei and Schito, 2022 [35]. We have decided on this solution for the reasons previously described.

### 3.1. Antimicrobial Properties of Carbon Nanotubes

#### 3.1.1. Mechanisms of Action and Influencing Factors

It has been reported that CNTs, especially SWCNTs and MWCNTs, possess excellent antibacterial and antifungal activity. Several mechanisms have been suggested to justify the CNTs’ toxicity against bacteria. Figure 5A shows the most recognized antimicrobial mechanisms [90].

Kang et al. reported, for the first time in 2007, that SWCNTs demonstrated strong antibacterial activity against *Escherichia coli*, by cell membrane impairment via direct contact, thus inhibiting 80% of bacterial cells [26]. Another study by the same authors, using *E. coli*, on the gene expression analysis, disclosed that the potency of antibacterial activity of CNTs mainly depended on their size and that the a-specific disruption of the bacterial membrane was the main mechanism of their antibacterial effects [27]. In the same year, the same authors demonstrated that when exposed to MWCNTs, *E. coli* revealed significant oxidative stress (OS), complemented by cell membrane disruption, cell lysis, and release of intracellular contents [20]. Nagai and Toyokuni, who studied the differences and similarities between carbon nanotubes and asbestos fibres’ effects on cells during mesothelial carcinogenesis, also reported that the mechanism by which CNTs enter non-phagocytic cells was mainly based on the impairment of cell membrane by direct pore formation on their surface [91]. On the other hand, Kang et al. reported that the length of CNTs is pivotal for their interactions with bacterial membranes, where shorter tubes showed higher toxicity against bacteria [27]. Additionally, Aslan et al. remarked that shorter SWCNTs were more toxic to target cells due to the higher density of open tube ends [92]. A similar observation was also reported by Johnson, showing that nanotubes with smaller diameters manage to impair membranes of target cells via easier interactions with their surface. More specifically, concerning the inhibition of some bacteria and fungi, including *E. coli*, *Bacillus subtilis*, *Staphylococcus aureus*, *Pseudomonas aeruginosa*, and *Candida albicans* caused by exposure to SWNT, DWNT, and MWNT, it was demonstrated that their death was induced by the accumulation of CNTs following their entrapment onto the microbial cell surface [93,94]. Differently, Arias et al. observed that SWCNTs functionalized with -OH and -COOH groups used in buffered suspension caused *Salmonella* cell aggregation and the formation of SWCNTs-pathogens aggregates with an increasing size depending on SWCNTs concentrations [95]. Although the authors admitted that additional mechanisms for the toxicity of SWCNTs to bacterial cells could exist, this was assumed to be the main one. The toxicity of SWCNTs to bacterial cells was also dependent on a buffer of suspensions [95]. Many studies have confirmed that SWCNTs are more toxic to different pathogens than MWCNTs and cause their membrane disruption [20,28]. Kang et al. found that while SWCNTs kill most *E. coli* bacterial cells after one hour of treatment, incubation with MWCNTs leaves bacteria alive [20,27]. The authors also evidenced that *E. coli* exposed to SWCNTs displayed higher levels of stress-related genes compared to MWCNT treatments. On the contrary, Young et al. demonstrated that the MWCNTs are more toxic than SWCNTs to *E. coli* [96]. Also, Saleemi et al., who have assessed the antimicrobial effects of DWCNTs and MWCNTs against *S. aureus*, *P. aeruginosa*, *Klebsiella pneumoniae*, and *C. albicans*, demonstrated that non-covalently dispersed MWCNTs exhibited higher antimicrobial activity than DWCNTs [97]. Despite the contradictory reports, which need further investigations, CNTs are very competitive compared to other nanomaterials in inhibiting a broad range of microorganisms, including both Gram-positive and Gram-negative species and fungi, such as *S. aureus*, *E. coli*, *Enterococcus faecalis*, *Lactobacillus acidophilus*, and *Bifidobacterium adolescentis* [20,26,28]. As reported in Figure 5B, several key factors can affect the antimicrobial efficacy of CNTs, including their composition, geometry, surface modification, intrinsic properties, and electronic structure [90]. In this regard, metallic SWCNTs possess higher antimicrobial effects than semiconducting CNTs due to their capability to cause oxidation of pathogens intracellular components [98]. Anyway, the antibacterial activity of CNTs may also depend on the species and morphology of microorganisms [28,99], the mechanical properties of cell surfaces [99], and their growth state (planktonic or sessile) (Figure 5B) [100]. Gram-positive bacteria, such as *B. subtilis* and *S. aureus*, are more susceptible to CNTs piercing due to their softer surface, thus leading to higher bacteria death rates [28,99]. Chen et al. demonstrated that CNTs work better against spherical-shaped pathogens than rod-shaped ones [28]. Moreover, as observed for traditional or innovative antibiotics, when microorganisms protect themselves within the structure of biofilm, they are difficult to reach, thus becoming less susceptible also to the effects of CNTs [100]. Anyway, it has been reported that longer immobilized CNTs prevent bacterial settlement and biofilm growth [101], while vertically aligned arrays of carbon nanotubes (VACNTs), consisting of tubes much smaller than the usual size of a bacterial cell, can reduce biofilm formation [102]. However, since the overall mechanisms by which CNTs prevent biofilm establishment and maturation have not yet been fully clarified, more extensive research is required to design robust and effective CNT-based coating materials to impede biofilm formation [90]

#### 3.1.2. Most Relevant Case Studies on the Antimicrobial Effects of Non-Modified CNTs

Table 1 below collects some relevant case studies reported over the years about the antimicrobial properties of non-modified CNTs.

Collectively, the results from studies reported in Table 1 established that both pristine SWCNTs and MWCNTs prepared by different methods possess moderate to remarkable antimicrobial and microbicide effects (1.5–250 µg/mL) against several species of Gram-positive and Gram-negative bacteria and *C. albicans*. As shown above in Figure 5A, most studies have confirmed that CNTs inhibit pathogens mainly by causing irreversible damage to their outer membrane. Upon aggregation on their surface via electrostatic interactions, severe impairments of microbial cell integrity occur by length-dependent wrapping and diameter-dependent piercing, determining cell lysis and lethal release of intracellular contents. Additionally, CNTs have shown the capability to impede the adhesion of bacteria on surfaces, thus paving the way for their promising application to prevent biofilm formation. Generally, the antimicrobial activity of MWCNTs was higher than that of DWCNTs but lower than that of SWCNTs.

#### 3.1.3. Most Relevant Case Studies on the Antimicrobial Effects of Modified CNTs

Several other studies investigating the antimicrobial properties of SWCNTs and MWCNTs, have been focused specifically on assessing the effect of their functionalization with different chemical groups and/or their modification by combination with metal nanoparticles (MNPs), polymers, antimicrobial peptides (AMPs), dendrimers, antibodies, etc., on their antimicrobial ability toward various bacterial and fungi strains. In this regard, Figure 6 shows the main synergistic effects observed by associating AMPs, MNPs, and polymers with CNTs [90].

In the following, a brief description of the most relevant results observed using modified CNTs is reported.

##### Modified SWCNTs

CNTs can be functionalized by both covalent and non-covalent methods. The functionalization of the surface of CNTs could have different goals, including the avoidance of desorption processes and of undesired absorption of molecules from the biological media, as well as enhancing the effectiveness of their antimicrobial activity [107,108,109,110,111]. Correlations between the toxicity to bacteria and the physicochemical properties or the agglomeration status of functionalized SWCNTs (f-SWCNTs) were studied by Pasquini et al., discovering that no direct correlation was identified between the bacterial cytotoxicity and thermal, physicochemical, and structural properties of f-SWCNTs [112]. On the contrary, they found that the aggregation of nanoparticles had more incidence than the single chemical and physical properties of functional groups on the f-SWCNTs cytotoxicity [112]. Arias et al. functionalized SWCNTs with different groups and tested them in suspension to evaluate the possible increase in their interaction with pathogens and the effects on their antimicrobial activity [95]. Specifically, the authors studied the effects of various surface functional groups, including -NH_2_, -COOH, and -OH ones, on their antimicrobial effects against *S. aureus*, *B. subtilis*, and *Salmonella typhimurium*. SWCNTs bearing the cationic -NH_2_ group inhibited bacterial growth only at high concentrations. On the contrary, SWCNTs bearing anionic -COOH and neutral -OH groups reduced the viability of pathogens by 7 log CFU/g. To explain this fact, the authors assumed that the long carbon chain used to attach the NH_2_ groups to SWCNTs surface impedes the cylindrical shape of SWCNTs from being in close direct contact with microbial cell walls, thus probably reducing their inhibitory effects. On the contrary, -COOH and -OH groups were derived directly from the surface of SWCNTs, thus allowing the direct contact of bacteria with SWCNTs surface, thus sustaining and enhancing their antimicrobial potency. Several authors have reported on the antimicrobial activity of silver nanoparticles (AgNPs) and other metal oxides together with their inhibitory effects on the infections [113]. Chaudhari et al. observed that the antimicrobial properties of silver-coated SWCNTs on *S. aureus*, when applied in a skin model, can be modified with antimicrobial peptides (AMPs) [114]. In this regard, they observed that the proliferation of bacteria was reduced by 10^5^ CFU/g by silver-coated functionalized CNT after skin treatment [114]. Generally, AgNPs bind and penetrate the bacterial cell membrane, causing cell death by altering the permeability of the membrane and increasing ROS induction [115], while AMPs have shown antimicrobial effects on various fungi, bacteria, and viruses by the same mechanisms [116]. Therefore, the synergistic effects of AgNPs with AMPs increase the toxic effects of SWCNTs, thus allowing for the possible development of novel antimicrobial therapies [114]. The same authors tested their functionalized SWCNTs against *S. aureus*, *Streptococcus pyogenes*, *Salmonella enterica* serovar Typhimurium, and *E. coli* [117], finding that the conjugation led to a strong synergistic antibacterial effect of TP359 with SWCNTs-adsorbed AgNPs. Kumar et al. reported the excellent antibacterial potency of SWCNTs enriched with AgNPs inserted in cotton fabrics against *S. aureus* and *E. coli* [118]. Moreover, AgNPs embedded in the silica-coated SWCNT substrate inactivated *E. coli* growth better, with respect to AgNPs plasma-treated SWCNT substrates [119]. Eco-designed biohybrids based on liposomes containing cholesterol, mint–nano-silver, and carbon nanotubes demonstrated antioxidant and antimicrobial properties against *S. aureus*, *E. coli*, and *E. faecalis* [120]. Chang et al. used a simple one-step procedure to synthesize nanocomposites encompassing CNTs, graphene oxide, and AgNPs effective against *E. coli* and *S. aureus*, exhibiting high disinfection properties [121]. Further investigation proved that the nanocomposite of Chang and colleagues was able to induce O_2_-based OS on bacteria, which damaged cell membrane integrity, thus triggering cell death. SWCNTs coated with Ag-doped TiO_2_ nanoparticles demonstrated strong antibacterial activity against both *E. coli* and *S. aureus*, although *S. aureus* was more tolerant than *E. coli* under illumination by UV light [107]. Park et al. manufactured pegylated SWCNTs (pSWCNTs) covered AgNPs, and their antibacterial effects were investigated on foodborne pathogenic bacteria [122]. A significant reduction in proteins associated with bacterial biofilm formation and quorum sensing, as well as of proteins necessary for the conservation of cellular structure, were observed. This was associated with a decrease in cell motility in the foodborne pathogens that remained alive. Also, by a simple and low-cost one-pot synthetic procedure, Singh et al. produced a SWCNTs/Ag/PPy-based nanocomposite which succeed in fully inhibiting the growth of *S. aureus*, *P. aeruginosa*, *E. coli,* and *B. cereus* completely within 24 h treatment [123]. A new antimicrobial nano system made of mesoporous silica and AgNP-coated SWCNTs (SWCNTs@mSiO_2_-TSD@Ag), which demonstrated strong antimicrobial activity against multidrug-resistant (MDR) bacterial strains by impairing cell membrane and releasing of Ag ions, was projected by Zhu et al. [124]. Yun et al. manufactured CNTs-Ag and GO-Ag, which demonstrated antibacterial effects against both Gram-positive and Gram-negative species, even if the effects of CNTs-Ag were higher than those of GO-Ag nanocomposites [125]. Additionally, a carbon-Ag nanosized complex inhibited the microbial growth of methicillin-resistant *S. aureus*, *K. pneumoniae*, *Acinetobacter baumannii*, *Yersinia pestis,* and *Burkholderia cepacian* [126]. SWCNTs were also modified with natural enzymes, such as lysozyme (LSZ) succeeding in enhancing their antibacterial impact against different bacterial species, including *S. aureus* and *Micrococcus lysodeikticus*, by causing cell wall lysis via hydrolytic breck of the β-1, 4 bonds in peptidoglycan [127,128,129]. Chemical modifications of SWCNTs were mainly finalized to enhance their dissolution properties and chemical compatibility. On the contrary, the functionalization of SWCNTs with polymers, achieving deposited aggregates and membrane coatings, improved their dispersibility and solubility and increased the interfacial interaction to polymeric matrices in their composites, thus demonstrating high chemical stability and high toxicity toward the microbes. In contrast, pristine SWCNTs, polymer-modified SWCNTs are less expensive and provide an enlarged range of mechanical, degradation, and structural properties, thus being suitable for developing ideal antimicrobial nanomaterials. Several case studies have been available in the literature reporting on the antibacterial activity of these carbon-based nanocomposites. Aslan et al. ideated SWCNTs with poly-(lactic-co-glycolic acid) (PLGA) and explored their antibacterial effects against *S. epidermidis* and *E. coli*, observing a 98% viability reduction and a significant decrease in the metabolic activity of the bacteria [92]. On the other hand, the SWCNTs modified with polyvinyl-N-carbazole demonstrated a 90 and 94% inhibition of planktonic cells for *B. subtilis* and *E. coli*, respectively, while reducing their biofilm formation [130]. Nanocomposite deriving by refashioning SWCNTs with poly-(L-glutamic acid) and poly-(L-lysine) resulted in a 90% inhibition rate of *E. coli* and *S. epidermidis* [131]. Goodwin et al. synthesized a SWCNTs-poly-(vinyl alcohol) nanosized composite, which gradually inactivated *P. aeruginosa* cells depending on increasing concentrations of SWCNTs [132]. SWCNTs reformulated with porphyrin, which possessed appreciable antibacterial effects against *S. aureus* in the presence of visible light and used a tungsten-halogen lamp, were reported by Sah et al. [133]. The activity of these materials was based on a photochemical reaction, which ultimately transferred an electron to the atmospheric molecular oxygen to form ROS, which destructed the bacterial cell wall, thus driving bacterial death [133]. Also, SWCNTs covalently bound with polyamide membranes showed a 66% inactivation of bacteria, translating into a delay in membrane biofouling [134]. The well-known poly-(ethylene glycol) (PEG) coating agents, recognized for their high hydrophilicity and biocompatibility, were attached in their linear and branched form by Cajero-Zul et al. to the surface of CNTs, achieving a nanoplatform suitable to manufacture medical devices [135]. Even if SWCNTs-copolymer of star-shaped PEG and poly-(ε caprolactone) (PCL) did not possess antimicrobial activity, they demonstrated thermal and mechanical characteristics superior to those of polymeric matrix. On the contrary, the star-shaped PCL-PEG copolymer structure prevented bacterial growth [135].

##### Modified MWCNTs

MWCNTs with surface enriched with -COOH group caused a 30, 27, 20–40, and 15~50% reduction in the viability of bacteria, including *B. subtilis*, *P. aeruginosa*, *E. coli*, and *S. aureus* [136,137,138]. Chen et al. showed that MWCNTs possessing -OH, or -COOH functional groups showed a significant dose-dependent antimicrobial effect on microbes such as *E. coli*, *S. aureus*, *Enterococcus faecalis*, *Lactobacillus acidophilus*, and *B. adolescentis* [28]. Ding et al. observed similar properties against *Vibrio parahaemolyticus* [139], while Arias et al. discovered that MWCNTs functionalized with -COOH, -NH_2_, and -OH did not exert appreciable antimicrobial effects with respect to analogous SWCNTs [95]. Various studies evaluated the antimicrobial action of non-covalently dispersed DWCNTs and MWCNTs against *S. aureus*, *K. pneumoniae*, *P. aeruginosa*, and *C. albicans,* finding time- and concentration-dependent mechanisms [97]. MWCNTs decorated with ethanolamine suppressed the microorganism’s growth to a large extent, compared with non-modified MWCNTs [140]. Another study showed that MWCNTs modified with oxygen groups could increase their antimicrobial properties [141]. MWCNTs added with Ag NPs revealed significant antimicrobial performance, which is like the SWCMT counterpart. Specifically, silver/MWCNTs complex caused 94–99, 57, 100 and 70% inactivation of *S. epidermidis* and *E. coli*, *S. aureus*, *Sphingomonas* spp., and *Methylobacterium* spp., as well as *P. aeruginosa*, respectively [138,142,143]. Silver-coated-MWCNTs complexed with amphiphilic poly-(propylene imine) (PPIs) dendrimers inactivated >90% of *S. aureus*, *B. subtilis*, and *E. coli* strains [136]. When the silver/MWCNTs complex was functionalized with polymer colloids, it displayed strong antimicrobial effect against *S. aureus* and *E. coli* [144]. Also, silver sulphide (Ag_2_S) quantum dots associated to poly-(amidoamine) (PAMAMs)-grafted MWCNTs showed better antimicrobial activity than cadmium sulphide quantum dot-coated-MWCNTs resulting in a microbial growth inhibition by 56, 98 and 79% when *S. aureus*, *E. coli*, and *P. aeruginosa* were exposed [138]. Aiming at more favourable results, researchers also amalgamated MWCNTs with copper nanoparticles, thus decreasing the bacteria viability by 75% [145]. Zinc oxide evidenced strong antimicrobial activity against *E. coli* [146], titanium, and gold is observed to have remarkable microbial growth inhibition against *B. subtilis*, *K. pneumoniae*, *S. aureus*, *C. albicans*, *Streptococcus pneumoniae*, *Proteus vulgaris*, and *Shigella dysenteriae* [147]. Very recently, Kim et al. modified MWCNTs with ZnO NPs, obtaining an antibacterial nanocomposite that demonstrated remarkably antibacterial effects against *E. coli*, *P. aeruginosa*, *E. faecalis*, *S. aureus* (Figure 7B,D). When incorporated in a urinary catheter, it inhibited *E. coli* and *P. aeruginosa* biofilm formation by 53.42 and 56.44% after 120 h of incubation, respectively (Figure 7A,C) [148].

Moreover, titanium alloy-coated-MWCNTs combined with rifampicin inhibited the formation of biofilm for up to 5 days [149]. As in the case of SWCNTs, enzymes like chloroperoxidase (CPO) and laccase were impaled on the MWCNTs’ surface, resulting in nanocomplexes capable of reducing the viability of *S. aureus* and *E. coli* by 99%. It was observed that the laccase-MWCNTs were even capable of reducing the microbial growth and spore formation of *B. cereus* and *B. anthracis* by >99% [150]. Additionally, it was demonstrated that CPO in CPO-MWCNTs oxidated chloride into HOCl by H_2_O_2_ in acidic conditions, driving the formation of singlet oxygen. It and HOCl functioned as strong oxidants capable of inhibiting, controlling, or reducing microbial growth of *E. coli* and *S. aureus* [150]. On the contrary, it was understood that the main antimicrobial agent in the laccase-methyl syringate (MS) system was the hydroxyl radical produced mainly by the Haber–Weiss reaction starting from H_2_O_2_ and superoxide radical, which instigated detrimental and irreversible impairments in the bacterial cells leading to their destruction [150]. When MWCNTs merged with polymers, the antimicrobial activity was maintained. The bacterial growth of *E. coli*, *B. subtilis*, and *S. aureus* was inhibited by 87% and 97% using MWCNTs modified with amphiphilic PPIs dendrimer synthesized by Murugan et al. [136]. Similarly, Neelgund et al. formulated MWCNTs with an aromatic polyamide dendrimer, which exerted good antimicrobial effects on *P. aeruginosa* (65.2%) and *E. coli* by inhibiting their growth by 65 and 73%, respectively [137]. Also, MWCNTs functionalized with PAMAMs inhibited the growth of several selected bacteria in an NT concentration-dependent way [138], as also reported by Goodwin et al. for MWCNT-poly (vinyl alcohol) nanocomposites [132]. A current trend is increasingly driving the interest of researchers toward the evaluation of the antimicrobial activity of MWCNT-based chitosan hydrogels due to the biocompatibility of the hydrogel-based materials. In this regard, a robust antimicrobial activity of MWCNT-based chitosan hydrogels was confirmed against *S. aureus*, *E. coli*, and *C. tropicalis* [151], while Mohamed et al. informed that MWCNT-based chitosan possessed a broad-spectrum antimicrobial activity [152]. The following tables, Table 2 and Table 3, schematically collect the most relevant case studies concerning the antimicrobial effects of modified SWCNTs and MWCNTs, respectively.

From data reported in Table 2, we noted that the functionalization of SWCNTs with oxygenated groups (-OH, -COOH) translated to more enhanced bactericidal activity, while the improvement derived by amine group insertion was inferior. Combination of SWCNTs in nanocomposites led to remarkable antibacterial effects (>90% bacteria inactivation) when PLGA, PVK, and PLG/PLL polymers were used, while the antibacterial activity was moderate when PEG/poly-ε-caprolactone or polyamide polymers were utilized. Interestingly, SWCNTs-PVK induced 90% and 94% of *B. subtilis* and *E. coli* inhibition in the planktonic cells and a significant reduction of biofilm formation, thus being promising to counteract this difficult-to-treat form of pathogens resistance, causing orthopaedic implants-associated severe infections. Concerning modified MWCNTs, they generally demonstrated a time- and concentration-dependent antimicrobial activity toward both bacteria and fungi. In some cases, modified MWCNTs did not pierce the pathogen’s membrane or the non-penetrated cells, but they wrapped around their surface, causing severe injury without lysis. Anyway, in most cases, including MWCNTs modified with gold (Au) NPs, upon contact via electrostatic interactions, MWCNTs inhibited microbes by damaging their external membrane up to its disruption and cell penetration. Functionalization to provide MWCNTs-OH and -COOH materials did not significantly induce antimicrobial activity or caused a moderate to scarce inactivation of 15–50% for *B. subtilis*, *E. coli*, *P. aeruginosa,* and *S. aureus.* Better antibacterial activity was observed against *S. aureus*, *B. subtilis,* and *E. coli* when MWCNTs were merged with PPI (87–97% inactivation). Differently, limited antibacterial effects were displayed by MWCNTs against *E. coli*, *P. aeruginosa,* and *S. aureus* when modified with an aromatic polyamide dendrimer (36–73% bacteria inactivation) or a PAMAM one (23–60% bacteria inactivation). Generally, modified CNTs exhibited higher antimicrobial activity against Gram-positive bacteria due to their softer surface and lower resistance to piercing and damage.

## 4. Impediments to the Extensive Application of CNTs: Toxicity Issues

Since the invention of CNTs in 1991 [17], researchers have not thought about the possible negative effects of their use and contact with humans and animals or about their environmental impact, and so they were not investigated [159]. This attitude was mostly due to the limited production processes available at that time, which forced researchers to generate CNTs only on a laboratory level and at a high cost [160]. Only when novel methods, such as CVD, made their production possible on a massive scale since 2000 did the impact of CNTs on health become progressively studied through toxicological studies [161]. It was mainly their shape, like that of asbestos fibres, that prompted the very first health warnings. In fact, asbestos fibres were already identified as material causing inflammation and leading to lung cancer, which starts when macrophage cells try to absorb the needle-like asbestos fibres but fail because they are too long, thus causing the activation of the so-called giant cells. The constant activation of these cells can drive to the establishment of a nodular tissue (granuloma), which may translate into mesothelioma over a long latency period of thirty to forty years. On the other hand, since a huge range of CNTs exists, their possible negative biological effects depend mainly on their shape. To check this assumption, most studies included and include experiments on cells and animal models using rats or mice. Animals were exposed to CNTs and inhaled them through the nose or were subjected to fibres application directly to their lungs or thoraxes. The surface characteristics of CNTs also affect the possible biological outcomes deriving from cells and animals’ exposure to these materials. An injection of 400 μg/mouse of water-soluble SWCNTs has been shown to be quickly expelled through the kidneys with a half-life of 3 h. Similarly, other researchers reported that hydroxylated SWCNTs, regardless of the form of administration, were quickly removed from the body through the urine [162], thus establishing that the contact of the body with these non-bio persistent CNTs could be very short and not worrying [159].

### 4.1. Introduction to Environmental and Human Safety

Generally, the results from some reported case studies concerning the impact of exposure of humans and the environment to CNTS are contradictory. Indeed, fruit fly larvae nourished with a CNTs-rich diet developed normally [35], while, despite the fish foetus appearing normal, CNTs delayed embryo development in zebrafish [163]. Inflammation of the lung like that caused by asbestos fibres was perceived for a few months in mice exposed to CNTs [164]. Certain human tumour cells proliferated more rapidly when exposed to CNTs [165]. CNT-based solar cells need a cadmium-telluride mixture coating, which is extremely toxic, thus impeding the widespread use of such solar devices [165]. This establishes that frequently, coatings applied to the CNTs, rather than CNTs themselves, are environmentally dangerous. A highly worrying concern about the massive use of CNTs regards their slow biodegradation, which leads to the release of tubes in the environment, their passage into our food supply, and from food into our bodies, with consequences not fully clear until now. It must, however, be considered that CNT applications in electronics are unlikely to be very risky because of the small volumes involved. Collectively, more inspections and regulation measures are necessary in this field, and it would be suggestable to treat CNTs as a new chemical material rather than as an allotrope of inert carbon [165].

#### Other Hurdles Are Close to Solution

Although the potential of CNTs is huge, obstacles to their production and application on a large scale endure. First, CNT-based devices were manufactured at prohibitive costs for typical consumers [83]. For years, both researchers in industry and academic laboratories have been making efforts to develop automated systems for growing CNTs endowed with uniform and predictable properties, thus making it possible to reduce their unclear toxicity. Indeed, until they realize a method of ideally structurally perfect CNT production on a huge scale, most of their possible applications will remain in the research laboratory, and silicon will manage most technologies, including the computing one [83]. Fortunately, nowadays, this era is closer than we might think. Very recently, a carbon copilot (CARCO), an artificial intelligence (AI)-driven platform that integrates transformer-based language models, robotic chemical vapor deposition (CVD), and data-driven machine learning models, has been developed by Li et al. [166]. Employing CARCO, the authors found a new titanium-platinum bimetallic catalyst for high-density horizontally aligned carbon nanotube (HACNT) array synthesis. This catalyst outperformed traditional ones and was treasured in millions of virtual experiments; an unprecedented 56% precision in synthesizing predetermined densities of HACNT arrays was achieved [166].

### 4.2. In Vitro and In Vivo Studies

#### 4.2.1. In Vitro Studies

Over the years, increasing in vitro investigations about the cytotoxicity and genotoxicity of CNTs have been reported. Following is a collection of early studies developed in the years 2003–2011. The cytotoxic effect of titanium oxide (TiO_2_) nanoparticles (NPs) and MWCNTs was studied in A549 human pneumocytes by Simon-Deckers et al., who evaluated cell viability and intracellular accumulation of both nanomaterials and compared results [167]. Both CNTs and TiO_2_ NPs were capable of entering cells and passing into the cytoplasm. Anyway, TiO_2_ NPs showed lower toxicity. Moreover, it was found that neither the presence of metallic impurities nor the length of CNTs influenced their cytotoxicity [167]. The capability of CNTs to penetrate the cellular membrane of both NR8383 rat macrophages and human pneumocytes A549, thus causing alteration of physiology and cellular functions, was reported by Pulskamp et al. [168]. Disfunctions are derived from a dose-dependent increase in intracellular reactive oxygen species (ROS) and from a decrease in the potential of the mitochondrial membrane in both rat NR8383 cells and human A549 lung ones. Interestingly, CNTs with deprived or with reduced metal content due to purification work-up demonstrated few or none of these undesired effects [168]. Collectively, several researchers proposed that the main response to the negative biological effects of commercial CNTs could be metal impurities. Moreover, it was reported that the cytotoxic effects of CNTs could heavily depend on their different degrees of agglomeration. Studies on human MSTO-211H cells revealed that the dispersion of CNTs in surfactant was less toxic compared to agglomerated CNTs, especially when CNTs agglomerated in string forms, which are more voluminous, more rigid, and more solid [169]. In this regard, Montero et al. studied the possible per se toxicity of different surfactants, finding that Pluronic F127 was less toxic [170]. MWCNTs, as well as MWCNTs coated with Pluronic F127, were tested on human keratinocytes, evidencing that the uncoated MWCNTs were found to be more cytotoxic [170]. Based on results from several studies, which confirmed the capability of CNTs to cross cell membranes [167,171], the intracellular distribution of modified SWCNTs was analysed in human fibroblasts 3T6 and murine 3T3 cells by Pantarotto et al. [171]. The results of this study demonstrated that functionalized CNTs could cross the cytoplasmic membrane and accumulate in cytosol or reach the nucleus without toxicity for the cells up to 10 µM concentrations [171]. Pantarotto et al. and Manna et al. investigated the toxicity of SWCNTs in HaCaT, HeLa, H1299, and A549 cells at different concentrations and exposure timings, finding a major loss of cell viability in keratinocytes (HaCaT), probably due to the activation of keratinocyte-specific transcription factor NF-κB by increased oxidative stress (OS) [171,172]. Anyway, Shvedova et al. have already explained that the unpleasant consequences demonstrated by SWCNTs in HaCaT and bronchial epithelial (BEAS-2B) cells, including alterations in cell ultrastructure and morphology, loss of cellular integrity and cellular apoptosis, OS, and depletion of antioxidants in the cells is specifically based on the alteration of certain genes following exposure to SWCNTs. Additionally, the authors declared that exposure to SWCNTs can trigger cutaneous and pulmonary toxicity [173]. Later, the same authors administered mouse macrophages (RAW 264.7) with different concentrations of SWCNTs, finding that even well-dispersed SWCNTs penetrated through the alveolar epithelium, caused interstitial fibrosis and alveolar wall thickening [174]. The production of TGF-β1 was triggered similarly to the effect of zymosan. Anyway, neither oxidative response nor nitric oxide production or cellular apoptosis was induced by CNTs [174]. The uptake of SWCNTs enriched with fluorescein-isothiocyanate by lymphocytes and macrophages reported by Dumortier et al. [175], upon in vitro experiments, did not lead to any changes in cell viability. These findings confirmed previous reports on in vitro cytotoxicity studies using human dermal fibroblasts exposed to modified SWCNTs, which demonstrated toxic effects lower than those observed using not modified SWCNTs [176]. Unexpectedly, while streptavidin-conjugated SWCNTs exhibited low toxicity in HL60 cells, the SWCNT–biotin–streptavidin complex caused cell death [177]. However, despite the functionalization of SWCNTs generally enhancing their potential by reducing their toxic effects, a general increase in toxicity was observed for the MWCNTs bearing carbonyl (C=O), carboxyl (COOH), and/or hydroxyl (OH) groups [178]. Moreover, indirect cytotoxicity of CNTs was reported deriving from their capability to stimulate immune-mediated cytotoxicity in various human cells, even at low concentrations (0.001–0.1 mg/mL). In this regard, it was reported that CNTs at lower concentrations may stimulate the secretion of cytokines with consequent activation of lymphocytes and upregulation of the NF-κB expression in immune cells [179]. The effect of SWCNTs on human epidermal keratinocytes (HEK293T) was investigated by Cui et al. using several techniques [180]. A dose- and time-dependent decrease in cell proliferation and adhesive ability was observed, with the formation of nodular structures, induction of G1 arrest, and apoptosis. SWCNTs (10–20 nm in diameter) showed higher toxicity than MWCNTs of the same dimensions, while fullerenes did not show any cytotoxicity when they were administered to alveolar macrophages in guinea pigs. These results suggested that the geometric structures of carbon nanomaterials are discriminant for cytotoxicity [181]. Pacurari et al. [182] demonstrated that SWCNTs induced ROS generation, increased cell death, and enhanced DNA damage and H2AX phosphorylation due to their fibrous characteristics involved in ROS generation. The DNA damage occurring upon exposure of lung V79 fibroblasts to acid-purified SWCNTs was confirmed by Kisin et al. [183]. According to Francis et al., Patapovich found that, while iron-rich SWCNTs activated the macrophages and catalysed the transformation of extracellular O^2–^ into detrimental hydroxyl radicals, thus encouraging IL-6 production, and causing OS and inflammation, the iron-deprived SWCNTs stimulated TGF-β production, thus promoting apoptosis [184]. MWCNTs were found capable of causing the release of proinflammatory cytokines (TNF-α) in rat peritoneal macrophages [185]. The cytotoxic effects of MWCNTs in rat and mouse peritoneal macrophages (J774.1) were assessed by Muller et al. and Hirano et al. They found higher cytotoxicity of CNTs with respect to that of crocidolite in mouse macrophages due to the action of MWCNTs with MARCO (macrophage receptor with collagenous structure), resulting in the disruption of the plasma membrane [185,186]. Human embryonic kidney 293 cells (HEK-293) exposed for prolonged times to high concentrations of MWCNTs showed reduced viability, a significant increase in IL-8, and altered expression of several proteins, supported by an increasing penetration of MWCNTs via cell membrane over time [187,188]. Also, MWCNTs induced cytotoxicity through GSH depletion, ROS generation, OS, cell inflammation, membrane leakage, lipid peroxidation, and protein release [189]. When human aortic endothelial cells were treated with CNTs, an increase in the messenger RNA of MCP-1, VCAM-1, and IL-8 was observed [184], while when human skin fibroblasts (HSF42) were exposed to MWCNTs and multi-walled carbon nano-onions (MWCNO), inhibition of the cellular cycle and an increase in apoptosis and necrosis, were detected [190]. Several intracellular signalling pathways were altered after exposure to CNTs, depending on the material tested and the dose, establishing that MWCNTs are more toxic than MWCNO by an interferon and p38/ERK-MAPK mediated toxicity. Soto et al., using chrysotile as a positive control, explained that the cellular response to MWCNTs aggregates was like that observed for the chrysotile-induced response [191]. Similarly, Murr et al. showed that SWCNTs aggregates, two types of MWCNTs aggregates, and the chrysotile aggregate provoked cell death starting at 2.5 µg/mL with similar cytotoxic response in mouse alveolar macrophages [192]. Also, Bottini et al. compared the time- and dose-dependent toxicity to T-lymphocytes caused by not modified MWCNTs- and oxidized MWCNTs finding that the latter induced significant apoptosis, being more toxic than hydrophobic pristine MWCNTs, which can result toxic only at high concentrations [193]. The studies reported on the cytotoxicity of CNTs are summarized in Table 4.

#### 4.2.2. In Vivo Studies: Pulmonary Toxicity

Over the years, experts in the field have developed increasing specialization in devising synthetic and characterization methods for obtaining improved CNTs in large scale. At the same time, the application possibilities of CNTs have grown exponentially to the point of making more studies on their possible cytotoxicity and genotoxicity imperative, also in vivo. The almost daily use of CNTs in novel nanotube-based products greatly increased the possibility of contact with humans, animals, and the environment. CNTs can enter the body through various routes of exposure, including derma, mouth, and nose. Among the possible contact opportunities, CNTs can enter the respiratory airways of the workers and accumulate in the lungs first during manufacture. Also, since CNTs are used as fillers in food packaging products, they can reach the gastrointestinal tract (GIT) of the consumers. Several reports from in vivo studies on rodents have demonstrated that the ingestion of SWCNTs and MWCNTs is toxic. Based on these results, the possibility that CNTs could also be risky to humans has become an increasingly worrying reality, and confirming or refuting it is a necessity. Pulmonary toxicity of raw CNTs, containing iron impurities, acid-purified CNTs, and CarboLex CNTs-rich in nickel and yttrium impurities, was investigated by Lam et al. They dispersed all samples in the serum and in carbon black, or quartz particles were used as negative and positive controls [194]. The samples were administered intratracheally to mice at different concentrations. After 7 days, in the mice group treated with CNTs, epithelioid granulomas and interstitial inflammation were observed in a dose-dependent way, which augmented after 90 days. Moreover, peri-bronchial inflammation and necrosis were also discovered in the lungs of animals that were treated with CNTs via alveolar septa, thus establishing that CNTs are more toxic than carbon black and quartz, which led to negative health effects only in chronic inhalation exposures [194]. Pulmonary toxicity of pristine SWCNTs was assessed by Warheit et al., which used quartz particles as positive control and carbonyl iron particles as negative control, using rodents as animal models [195]. Rats were administered intratracheally with SWCNTs at 1 and 5 mg/kg. Upon administration of the latter dose, 15% of rats died within 24 h, probably due to the mechanical obstruction of the airways by the SWCNTs. Anyway, transient inflammation and cell damage were found in surviving animals, while multifocal granulomas, attributable to tissue reactions against the foreign body, were observed. These early findings confirmed the cytotoxic power of SWCNTs while demonstrating the huge necessity for more chronic study. Shvedova et al. showed that pharyngeal aspiration of SWCNTs induced anomalous pulmonary effects in C57BL/6 mice, which brought acute inflammation, rapid progressive fibrosis, granuloma, and alveolar wall thickening in the lungs [174]. These pathological lesions were associated with functional respiratory deficiencies and decreased bacterial clearance. To complete this worrying scenario, increased clinical marker values were detected in bronchoalveolar lavage (BAL) fluid, which endorsed that the SWCNTs were more toxic than crystalline silica [174]. Even higher inflammatory response, OS, collagen deposition, and fibrosis were observed in C57BL/6 mice when SWCNTs were inhaled rather than pharyngeal aspirated. Anyway, pharyngeal aspiration of SWCNTs provoked faster atherosclerotic plaque formation in ApoE mice [196]. Mutlu et al. administered intratracheally raw aggregated and highly dispersed SWCNTs in 1% Pluronic F 108NF to mice at a 40-µg dose, which was higher than the dose used by Shvedova et al. [174] to induce pulmonary fibrosis in mice [197]. Lung inflammation was induced by aggregated SWCNTs in PBS, while highly dispersed SWCNTs do not cause any adverse phenomena. Muller et al. reported the respiratory toxicity of both MWCNTs and ground MWCNTs suspended in sterile saline (0.9% NaCl), which was observed after 2 months of exposure in the form of pulmonary lesions characterized by collagen-rich granulomas, dose-related pulmonary fibrosis, and increased production of TNF-α [185]. The dangerousness of the inflammatory effect produced by MWCNTs was anyway halved with respect to asbestos and carbon black [185]. Mitchel et al. reported that the exposure of mice to MWCNTs at high doses caused immunosuppression after 14 days of whole-body inhalation. On the contrary, inflammation and granuloma formation were not observed, which was different from what was previously reported [198]. Later, Muller et al. administered intratracheally to rats (2 mg/rat) MWCNTs in the forms of both tubes heated to 600 °C and 2400 °C and ground MWCNTs heated at 2400 °C [199]. Observations revealed that the pulmonary toxicity of MWCNTs was less compared to those of ground MWCNTs, thus indicating that the toxicity of the tubes mostly depends on the imperfect sites in their carbon architecture. However, the group of Donaldson showed that the toxicity of ground MWCNTs may depend either on their larger dispersion in the lungs or on the effects of the released metals upon grinding [200]. Poland and his group reported that mice showed an asbestos-like pathogenic behaviour after exposition to long MWCNTs, which translated into inflammation and granuloma formation. Such a pathogenic scenario was also observed in the mice administered directly into the abdominal cavity with CNTs [32]. The additional exposure of lungs, with allergen-based inflammation, to SWCNTs caused increased pulmonary toxicity characterized by increased lung protein levels of T helper cytokines and chemokines, augmented the level of OS-related biomarkers and accessory allergen-specific IgG1 and IgE activity [201]. By light microscopic examination technique, Kobayashi et al. showed that MWCNT aggregates, accumulated in the lungs, were phagocytized by alveolar macrophages and remained up to 6 months post-exposure [202]. Granulomatous lesions or collagen accumulations were observed only in animals exposed to highly dispersed MWCNTs through intratracheal administration but not in those administered with MWCNTs by inhalation, thus demonstrating that MWCNTs produce pulmonary lesions, based on the route of administration and dose [203]. Also, Grubek-Jaworska et al. showed that four different commercial MWCNTs and SWCNTs, characterized by very low content of iron intratracheally administered, caused pneumonia with an interstitial non-specific focal reaction in guinea pigs, treated for 3 months [204]. Significantly increased concentrations of IL-8 in the bronchoalveolar lavage (BAL) fluids and augmented macrophages and eosinophils were instead caused by other types of CNTs. Additionally, carbon sediments were found mainly in the bronchioles, while the alveolar ducts and the alveoli were free. No granulomas were detected in lungs, while CNT aggregates and mechanical obstructions were found in certain airways of some animals. More recently, Francis et al. showed that the single exposure of rats to MWCNTs per se induced inflammation, epithelial cell membrane damage, and cell lysis, confirmed by augmented inflammation markers concentrations, including TNF-α, IL-4, LDH, WBC, and increased ALP activity [205]. Histopathological experiments revealed that dispersion of MWCNT in the lungs caused fibrosis and granuloma. Since occupational and intentional exposure to MWCNTs has exponentially increased over the years with large-scale production of CNTs, it is mandatory to investigate their toxicity upon prolonged exposure timings. For an easier understanding by the readers of the scenario of experiments carried out to accredit the possible pulmonary toxicity of CTNs, the early pulmonary toxicity studies discussed in this section on CNTs have been summarized in Table 5.

##### CNTs Cytotoxic Effects to Other Tissues

Initial animal studies reported previously in this section evidenced that exposure of rodents to different forms of CNTs can cause acute and chronic pulmonary inflammation, based on increasing OS and on the assumption of inflammatory effects of atherosclerosis and air pollution [174,185,194,195]. Based on these results, it was presumed that the toxic effects of CNTs could also affect other tissues. In this regard, some authors, such as Simenova and Erdely [206], as well as Li et al. [196], investigated the cardiovascular toxicity of CNTs in rodents by different ways of administration. The studies revealed that CNTs are capable of activating the blood cells via inflammatory markers release, which leads to adverse cardiovascular effects. Damage to the mitochondrial DNA of the aorta was observed in a stress-dependent and dose-dependent mode. The effects caused by CNTs may predispose to atherogenesis. An activated oxidative marker was found in the aorta and cardiac tissues of mice with high blood cholesterol levels exposed to SWCNTs, which also developed aortic DNA damage. Atherosclerotic plaques on the surface of the aorta and increased atherosclerotic lesions in the brachiocephalic arteries were observed in mice that had been exposed to SWCNTs [184,196,206]. Pietroiusti et al. [207], Bai et al. [208] and Philbrook et al. [209] studied the toxic effects of CNTs on the reproductive and developmental system. It was reported that functionalized CNTs administered at low doses were capable of triggering high resorption processes, progressive malformations in the survived infant rodents, and induced ROS generation in the placentas of exposed animals. A higher percentage of resorptions, as well as huge morphological defects and skeletal abnormalities in foetuses, were discovered in *Drosophila melanogaster* and CD-1 mice exposed to functionalized CNTs. CNTs accumulated in the testicles, causing OS, tissue damage, and alterations, which raised concerns about possible adverse effects on male fertility [207,208,209]. A collection of more recent case studies on in vitro and in vivo CNT toxicity has been reported in Table 6, while Table 7 collects case studies on CMT toxicity in various organs. Table 6 and Table 7 already contain detailed explanations of observed outcomes and do not need additional discussion.

## 5. Possible Strategies to Moderate CNTs’ Toxic Effects: Future Perspective and Preventive Actions

CNTs cause a cascade of events that are noxious to human and animal body tissues. In this regard, Lettiero et al. and Moghimi et al. have reported that inhibitors of such events, achievable by CNTs surface modifications and functionalization, could prevent the activation of the toxic system in the body tissues [232,233]. Generally, PEG-modified SWCNTs were not toxic and biocompatible to mice, achieving the longest blood circulation in 1 day and being fully eliminated from the vital tissues of mice in about 2 months [234]. Longer circulation time, minor uptake in the reticule-endothelial system, and reduced accumulation in the spleen and liver were observed for PEG-CNTs by Yang et al. [235]. Coating CNTs with C1q recombinant globular proteins and adding functional groups of different types to their side walls are other methods to minimize CNT toxicity [176,236]. Silva et al. reported that the onset of CNT toxicity is strictly correlated with the route of administration [237]. CNTs administered by inhalation provided no sign of inflammation after 1 day, but inflammation appeared after 21 days. On the contrary, instilled CNTs caused inflammation after 1 day of exposure, which disappeared after 21 days [238].

Adding catalytic horseradish peroxidase that promoted -COOH CNTs degradation in an acidic medium led to no toxic or inflammatory effects on the lungs of mice used for toxicity experiments.

Collectively, limiting the toxicity of CNTs remains a serious aspect of their safer use in various applications. Table 8 collects some relevant strategies proposed over the years up until now to mitigate potential harmful effects, which could arise from an extensive exposure to CNTs.

Proper modification of the surface of CNTs with biocompatible materials or selected molecules can enhance their dispersion in biological fluids and reduce toxicity. Functionalization can also influence cellular uptake and interactions [252,253]. One of the main causes of the hazardous effects of CNTs on the human and animal body is their poor water solubility. This is a problem that may be solved by CNT surface modification [254,255]. A current trend followed by scientists in the sector consists of searching for biocompatible substances that may be smeared with CNTs to improve their characteristics [256]. Much research has been done on the detrimental effects of CNTs on pulmonary systems, with confirming results on A549 cells [257,258]. By covering MWCNTs with curcumin, their propensity to induce OS, inflammation, and cell death was significantly reduced [243]. Wu et al. enriched MWCNTs-COOH and pegylated MWCNTs with oxaliplatin, a cisplatin-derived chemotherapeutic, by both their surface functionalization with the drug and its encapsulation in CNTs cavities [244]. The authors demonstrated that, in an aqueous environment, the entrapped oxaliplatin could easily exit MWCNT-COOH and MWCNT-polyethylene glycol (PEG) and that Pegylated MWCNTs were capable of an oxaliplatin-controlled release, thus augmenting its therapeutic effects.

Shorter and well-dispersed CNTs with a high specific surface area and robust transmembrane mobility showed lower toxicity compared to longer aggregates [245]. These substances can harm proteins and cellular components, leading to macrophage malfunction or even death [222]. Sohaebuddin et al. demonstrated that MWCNTs with large diameters provided minimal harm to lysosomes, while those with tiny diameters might create membrane instability [246]. Highly pure CNTs that do not contain residual metals or catalysts induce lower harmful effects [250]. Considerable progress has been achieved in the purification of CNTs [259]. Also, CNTs with structures that favour biodegradation over time could reduce their persistence in circulation, thus reducing tissue toxicity [260]. Concerning CNT-based delivery systems, the choice of alternative administration routes, including topical or transdermal ones, can minimize the direct exposure of sensitive organs or tissues to CNTs, thus reducing their systemic harmful impact.

Quercetin has good bioavailability and possesses formidable analgesic, antioxidant, anti-inflammatory, and anti-atherosclerotic properties [261]. Additionally, quercetin reduces reactive oxygen species (ROS) generation and increases the antioxidant enzyme activity [262]. It could be exploited as an adjuvant to prevent oxidative damage, inflammation, and immune-toxic effects of MWCNTs [251].

Despite the several strategies developed so far to limit the toxic outcomes of extensive use of CNTs to humans and the environment, it is mandatory to conduct preventive behaviour to limit the large diffusion of their possible cytotoxic impacts. Biocompatibility assays using appropriate in vitro and in vivo models are always necessary before a CNT’s widespread use. Knowing the individual biological responses to different types of CNTs is critical to assessing their specific risk [263]. Although a more specific regulation is urgently needed for these nonpareil nanomaterials, adhering to existing regulations and guidelines on nanomaterial safety is obligatory. The responsible development and use of CNTs could be assured only by ensuring compliance with regulatory standards [264]. Education training for workers who produce, handle, and apply CNTs, thus making them aware of potential risks which could derived by CNTs identified until now and implementing proper safety protocols can help to minimize harmless exposure. Concerning the environment, monitoring programs to evaluate the dispersion and potential impact of CNTs in workplaces and surrounding environments can help update risk management strategies. The invention and development of environmentally friendly or “green” synthesis procedures for creating CNTs as alternative options to the complicated machineries in use could reduce the usage of dangerous chemicals and toxic precursors and can lead to CNTs with fewer impurities and lower toxicity. Duraia et al. recently proposed a straightforward, simple method to generate CNTs on graphitic layers using corn seeds [265]. Alternative nanomaterials to CNTs with similar functionalities but lower toxicity risks should be considered. It is the case of graphene, which could be less hazardous than CNTs [266]. Indeed, as graphene is more robust than CNTs to immune cell destruction, thus limiting the possible release of hazardous compounds deriving from such degradation, it could be a promising option for CNTs. In particular, graphene oxide (GO) was less vulnerable to macrophage destruction than SWCNTs. Unfortunately, efficient standardized techniques to assess the toxicity of these nanomaterials are still challenged by difficulties and restrictions [267]. Permanent research and collaboration between scientists, engineers, and regulators are pivotal to developing safe routines for CNT production and use in various industries.

## 6. Regulatory Considerations

The National Institute for Occupational Safety and Health (NIOSH) is the most important federal agency in the United States, which conducts research and offers supervision for occupational safety and the consequences on human health from the extensive applications of and exposure to nanomaterials. Studies have revealed that nanoparticles (NPs) may pose a greater health risk than bulk materials due to their greater surface area/unit mass ratio [35]. In 2013, NIOSH published a Current Intelligence Bulletin detailing the potential hazards of CNTs and recommended the exposure limit for CNTs and nanosized fibres [268]. Subsequently, in October 2016, SWCNTs were registered within the European Union’s Registration, Evaluation, Authorization, and Restriction of Chemicals (REACH) regulations as compounds to be studied concerning their possible dangerous effects. As a direct consequence, the commercialization of SWCNT in UE was permitted up to 10 metric tons. REACH has a centralized database that records all chemicals produced or imported into the EU in quantities of one ton per year or more per supplier or importer. A chemical safety report must be submitted to REACH when the quantity of its production is ten tonnes or more. There have been frequent complaints about REACH not having taken into account the nano-specific characteristics of materials, but only the amount of their production. Fundamentally, substances of a specific interest may be subjected to REACH restrictions. If such restraints can be fulfilled by the European Chemicals Agency (ECHA), a thorough comment and consultation procedure must be carried out, and the approval of the Member States’ committee must be pursued for that substance. A Candidate List of Very High Concern Substances (SVHC) containing almost 40 substances was compiled in 2010 by ECHA, but no compound was a nanomaterial. This was mainly due to the high thresholds of production quantity for nanomaterials—1 ton per year or more to impose their registration and 10 tonnes per year or more to make it necessary to submit a chemical safety report. Chemical Regulation in the EU and the US (TSCA; Toxic Substances Control Act) used the chemical composition of a material to define regulation measures without considering particle size or nano-specific features. The problem of controlling carbon-based products has become particularly apparent, especially with the advent of their large-scale production. In REACH, carbon was initially considered a non-issue (“minimum risk owing to its intrinsic features”). This status of immunity was eliminated at the end of the year 2008 when a clear differentiation was made between carbon and carbon in its nano-scale forms. In this regard, C60-fullerene received a distinct CAS (Chemical Abstracts Service) registration number according to the worldwide norms of the chemical denomination in relation to carbon, carbon black, and graphite since these materials, despite their same molecular character, necessitate to be dissociated from each other. (Nano)tubes that would have a diameter smaller than 1 nm and a length exceeding 100 nm were not considered as nanomaterials until 2011 in EU. To address this potential omission, Recommendation 2011/696/EU included in the definition both fullerenes, graphene flakes, and single-wall carbon nanotubes, with one or more external dimensions below 1 nm, as nanomaterials [269]. Unlike the EU, the US introduced CNTs under the Toxic Substances Control Act to strict notification requirements in October 2010. Their production, import, or usage (also as a continuation of ongoing activity) shall be notified to the authority, which will decide whether to import or process the substance within 90 days. The only exclusion was agreed to CNTs stalwartly integrated into a matrix. Concerning health and safety on the job, in 2011, the European Agency of Safety at Work (EU-OSHA) compiled a literature review in the field of CNTs, dealing with occupational exposure, toxicology, and protective measures in the work environment [270]. It was also a compilation of advice and guidelines for technical occupational safety and personal protective equipment. The toxicological data available for SWCNTs and MWCNTs have also been summarized [270]. The purpose of the overview was to provide the Swedish Work Environment Authority with information and a basis for various types of measures [270]. In the United Kingdom—the Health and Safety Executive (HSE), referred to the “10 Recommendations” to be considered for health and safety purposes at the job, published by the Royal Society and the Royal Academy of Engineering, 25 years earlier on the application of nanotechnologies. Based on these rules, the HSE advised on precautionary measures and risk avoidance practices. In 2009, HSE considered CNTs as “extremely concerned substances”. The British Standards Institution (BSI) suggested an exposure limit of 0.01 fibres/mL, even if investigations to monitor this limit test were time-consuming [159]. In the US, the National Institute for Occupational Safety and Health (NIOSH) recommended an occupational exposure limit of less than 7 μg CNTs per m^3^ of air [159]. This exposure limit is the smallest concentration that can be correctly measured [271]. On the other hand, the exposure limit for 5 mg/m^3^ of airborne graphite dust set by the EU Occupational Safety and Health Administration (EU-OSHA) is about 1000 times higher [159]. It should also be stressed that dust particles are bigger and, therefore, heavier. Anyway, despite only a few types of CNTs being utilized, the precautionary principles should provide indications for health, safety, and risk limitations to be applied to all types of CNT-based existing products. Indeed, since CNT handling cannot be fully prevented, it should aim at a high standard of safety and control as well [272]. Although the determination of a correct occupational limit for CNT exposure is mainly contingent on exposure data and measurement technology, the US limits concentrations dictated for CNTs are considered by NIOSH high-risk to develop adverse effects on respiratory health. In this context, the promotion of research in this area is essential and mandatory. Also, measures to limit hazards by CNTs need improvement, and measures to maintain airborne CNTs to a minimum level and below the limits of exposure should be developed. Studies have demonstrated that high concentrations of needle-shaped, long, thin, and bio-persistent CNTs have a serious adverse effect on the respiratory tract. In this context, the mechanisms of action and the dose–effect relationship for as high a number of CNT types as possible need to be clarified to be capable of compiling a dangerous chart. Also, the life cycle of CNTs in the environment still has not been thoroughly assessed, and argumentative information on CNT ecotoxicity and credible exposure data are limited and need improvement to better assess their environmental risk. Although legislation on hazardous chemicals and regulations on occupational health and safety exists, it still does not offer specific requirements for handling CNTs. Even though specifications of REACH on the volume thresholds for registration (1 JT) and for the obligatory chemical safety report (10 JT) exist, they have been set at levels that do not include all CNT manufacturers and do not consider CNT-specific features. Although efforts are proceeding at international levels in occupational health and safety to set limits for occupational airborne exposure to CNTs, specific regulations are still not adopted. Analytical and detection methods need improvements through further research and development. Lawmakers, regulators, and environmental activists’ interest in nanomaterials has incessantly increased over the years, and the interrogation of the correct nanotechnology regulations is a daily challenge. The incessant development of nanomaterials and the daily application of nanotechnology have led to rigid regulatory standards in both animal and human manufacturing and utilization. The US Environmental Protection Agency (EPA) reported that CNTs encompass substantial regulatory concerns regarding toxicity and environmental safety [159]. The US National Environmental Policy Act (NEPA) has published a series of rules regarding the main factors for controlling nanomaterials, including quality control and manufacturing and product safety evaluation, like toxicity, clearance, biodistribution, and metabolism. In the following sections, we have briefly introduced the most relevant international organizations that work to develop and publish technical standards, which cover procedures for testing, classification, and characterization of several sorts of materials. Then, we have focused on the main standards concerning nanomaterials and CNTs published by the International Organization for Standardization (ISO) and the American Society for Testing and Materials (ASTM) International Organization.

### 6.1. International Standards Concerning Nanomaterials and CNTs

Different international organizations, including ISO (1946), ASTM, BSI (1901), IEC (1906), DIN (1917), ANSI (1918), and AFNOR (1926), were created to develop and publish key standards to chemically and physically characterize materials of every sort. In the following, we consider ISO and ASTM.

#### 6.1.1. ISO Standards

The International Organization for Standardization (ISO) brings global experts together to agree on the best way of operating in several sectors, including the preparation methods to achieve a product, its characterization, and how to manage a process. As one of the oldest non-governmental international organizations, ISO has enabled trade and cooperation between people and companies all over the world since 1946. The International Standards published by ISO must be approved by all ISO members and serve to make life easier, safer, and better (available online at https://www.iso.org/about, accessed 15 April 2025). ISO has provided several standards, guidelines, methods, and requirements concerning differently structured nanomaterials, CNTs, and CNT-containing nanocomposites. The most recent ones concerning CNTs and CNT-containing nanocomposites are available online at https://www.iso.org/search.html?PROD_isoorg_en%5Bquery%5D=carbon%20nanotubes (accessed on 15 April 2025) and have been included in the following Table 9.

#### 6.1.2. ASTM Standards

American Society for Testing and Materials (ASTM) international organization is a nonprofit organization that develops and publishes approximately 12,000 technical standards covering the procedures for testing and classification of materials of every sort. It encompasses 30,000 members from 135 countries. ASTM also serves as the administrator for the US TAGs (United States Technical Advisory Group) to an enormous amount of ISO/TCs (International Organization for Standardization/Technical Committee) and to their subcommittees. Specifically, ASTM Committee E56 on Nanotechnology was formed in 2005. Nowadays, it comprises one hundred and eighty members from twenty-two countries and is formed by nine sub-committees with twenty-eight standards developed, which are available in Volume 14.02 of the Annual Book of ASTM standards. This Committee addresses issues related to standards and guidance materials for nanotechnology and nanomaterials, as well as the coordination of existing ASTM standardization related to nanotechnology needs. These international standards address topics including nano-enabled consumer products, physical and chemical characterization, and intellectual property issues (available online at https://www.astm.org/membership-participation/technical-committees/committee-e56, accessed on 15 April 2025). As an example, some key standards comprise the following:E2490 Standard Guide for Measurement of Particle Size Distribution of Nanomaterials in Suspension by Photon Correlation Spectroscopy (PCS);E2865 Standard Guide for Measurement of Electrophoretic Mobility and Zeta Potential of Nanosized Biological Materials;E2864 Standard Test Method for Measurement of Airborne Metal and Metal Oxide Nanoparticle Surface Area Concentration in Inhalation Exposure Chambers;E2524 Standard Test Method for Analysis of Haemolytic Properties of Nanoparticles;E2526 Standard Test Method for Evaluation of Cytotoxicity of Nanoparticulate Materials in Porcine Kidney Cells and Human Hepatocarcinoma Cells;E2535 Standard Guide for Handling Unbound Engineered Nanoscale Particles in Occupational Settings;E3025 Standard Guide for Tiered Approach to Detection and Characterization of Silver Nanomaterials in Textiles;E3275-21 Standard Guide for Visualization and Identification of Nanomaterials in Biological and Nonbiological Matrices Using Darkfield Microscopy/Hyperspectral Imaging (DFM/HSI) Analysis;E3172-18 Standard Guide for Reporting Production Information and Data for Nano-Objects;D8208-19 Standard Practice for Collection of Non-Fibrous Nanoparticles Using a Nanoparticle Respiratory Deposition (NRD) Sampler.

Anyway, we have only found a recent document specifically concerning CNTs: ASTM-D8526. This is the Standard Test Method for Analytical Procedure Using Transmission Electron Microscopy for the Determination of the Concentration of Carbon Nanotubes and Carbon Nanotube-containing Particles in Ambient Atmosphere (available online at https://www.document-center.com/standards/show/ASTM-D8526, accessed on 15 April 2025).

Despite ASTM also serving as the administrator to a huge amount of ISO/TCs and to their sub-committees, and its involvement in the development of standards about nanomaterials, our findings have evidenced that its specific interest in CNTs is extremely limited compared to that of ISO, thus confirming the great gap between the work and opinion of the different organizations worldwide about CNTs, which limit their regulation and hamper their safer production and utilization.

## 7. Conclusions and Perspectives

In this article, the main knowledge and findings gathered so far about CNTs have been reviewed, and we find that these tube-like nanomaterials, which are made only of carbon atoms or functionalized carbonaceous tubes, possess nonpareil properties, which enable them to be applied in several sectors with excellent results and fantastic improvements. Particular attention has been paid to their antimicrobial effects and toxic aspects on humans, animals, and the environment. These characteristics were confirmed by several case studies reporting in vitro and in vivo experiments developed over the years, which were inserted in reader-friendly tables. First, by a survey on the Scopus database, we have shown that research interest for CNTs in the years 2010–2025 has remained constant, with peaks of publications in 2016 and 2017. This is a feature probably due to the passage of CNTs’ larger-scale production and the development of more efficient production procedures. Our specific research about their biomedical applications has revealed that both variants of CNTs, referred to as SWCNTs and MWCNTs, find application in numerous areas of nanomedicine. Also, concerning the more restricted field of their applicability as antimicrobial agents, our study has highlighted that these carbon-based nanomaterials (CNMs) are very promising as novel antimicrobial agents due to their extensively reported detrimental effects against both Gram-positive and Gram-negative bacteria, fungi, and biofilm. Due to their inhibiting effects on biofilm and excellent mechanical properties, CNTs could represent new optional ingredients for orthopaedical implant construction. Unfortunately, although the results from toxicological tests are often contrasting, several early and recent investigations have unveiled that CNTs could be toxic to humans, animals, and the environment. CNT toxicity is mainly due to their surface construct, nanosized diameters, needle-like structures, and residual metal impurities, and it depends on concentration and exposure times. Initially, the CNTs’ possible toxicity was disregarded because they were made mainly of normally harmless carbon. Anyway, investigations prompted by the similarity of the needle-like CNT fibres with those of asbestos evidenced that they can be remarkably toxic and genotoxic to several cells, organs, and animal models. As with asbestos, the higher toxicity has resulted in pulmonary injuries, which is particularly worrying for CNT manufacturers who have long-time exposure to them and are at risk of inhaling their fibres. Toxicity to the cardiovascular system, as well as immune and reproductive systems, embryos, and neurotoxicity, have also been reported. Fortunately, several strategies, including preventive behaviours, which have been discussed here, have already been developed to limit the CNTs’ toxic effects, but more and incessant studies should be implemented to make their production and employment safer. This study has also shown that despite the significant advancements in terms of regulations concerning CNT production and possible applications, as well as about health and safety at the production sites, several unsolved regulatory issues remain, which need urgent solutions. The incessant development of nanomaterials and CNTs and their daily application has led to rigid regulatory standards developed by different international organizations in both animal and human manufacturing and utilization, but standards and guidelines specific to CNTs are still very limited. Substantial regulatory concerns regarding CNT toxicity and environmental safety remain. Future perspectives should consider improved research finalized to the development of standardized protocols for toxicity assessment, which would allow for more dependable and comparable outcomes. More sophisticated in vitro and in vivo models that closely mimic human biological systems are needed. These will enable more precise forecasts of the behaviour and toxicity of CNTs in real-world situations, as well as the predictions of the long-term health effects resulting from exposure to CNTs. Surely, incessant CNT safety evaluations, including toxicity, clearance, biodistribution, and metabolism, can help in CNT inspection and in reducing conflicting results, thus allowing for a more reliable toxicological assessment. Additionally, the practical challenges of scaling up CNT-based technologies still remain unsolved. Low-cost, eco-friendly, and sustainable synthetic methods for preparing CNTs with precise and pure architecture on a macroscopic scale are necessary. Currently, the fabrication of CNTs requires expensive and energy-intensive professional methods like CVD or AD, generates a combination of different kinds of CNTs, and necessitates long secondary purifying procedures. High-purity, homogenous CNTs can be challenging to produce, thus resulting in harmful materials to human health. Despite these drawbacks, current research also uses artificial intelligence (AI) and strives to resolve these problems and realize the full potential of CNTs in various applications.

In fact, it must be recognized that CNTs, despite their still unclear toxicity and insufficient regulation, are ubiquitously exploitable for improving the quality of human life. SWCNT catalyst support can enhance the selectivity of copper (Cu)-based electrocatalysts in the CO_2_ conversion into valuable chemicals, which is a key strategy for achieving net-zero carbon emissions. Moreover, CNTs can be used as co-catalysts, as well as catalyst supports, for enhanced fuel cell applications and water splitting, which are excellent techniques for alternative clean energy systems due to zero carbon dioxide release, recyclability, and high gravimetric energy densities. Additionally, CNT-based single-atom catalysts (SACs) for electrochemical nitrogen reduction reactions (NRR) have received increasing attention due to their sustainable, efficient, and green advantages. In this regard, we hope that this review could stimulate further research on these materials, which are aimed at allowing their safer large-scale production and utilization.

## Figures and Tables

**Figure 1 nanomaterials-15-00633-f001:**
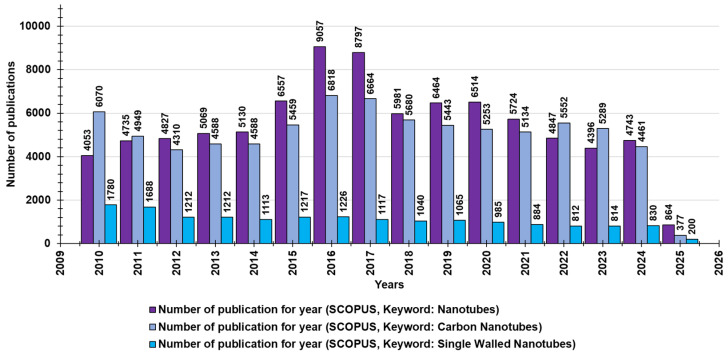
The number of publications per year during the last 15 years, according to Scopus, concerning nanotubes (purple bars), carbon nanotubes (light purple bars), and SWNTs (blue bars).

**Figure 2 nanomaterials-15-00633-f002:**
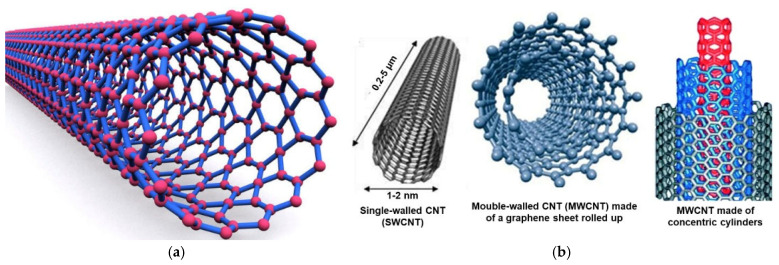
Structure of CNTs (**a**); structures of a SWCNT (left), of two different shapes of a MWCNT (center and right) (**b**). Specifically, an MWCNT made of a single graphene sheet rolled up (center) and an MWCNT made of more than two graphene cylinders with one inside the other (right). These images, by an unknown author, are licensed under CC BY and have been reproduced from our previous work [35].

**Figure 3 nanomaterials-15-00633-f003:**
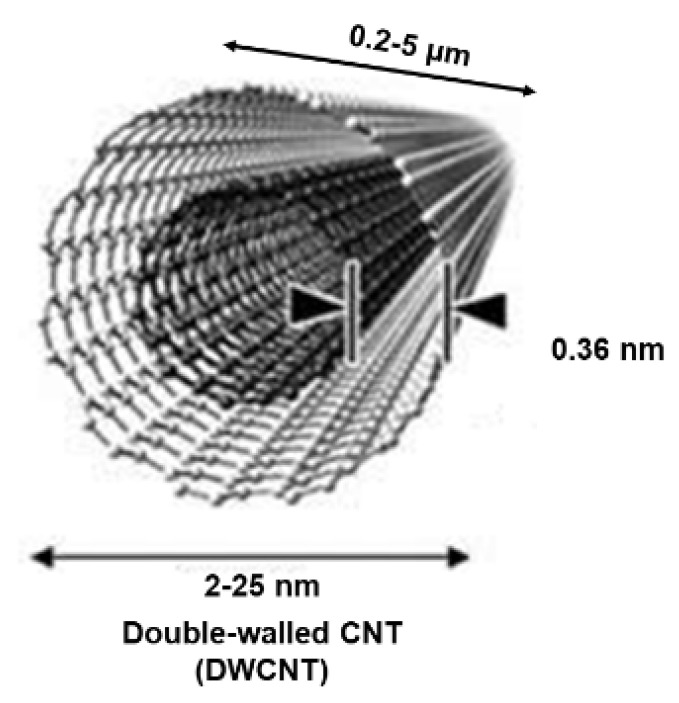
Structure of a DWCNT. This image by an unknown author is licensed under CC BY and was reproduced from our previous work [35].

**Figure 4 nanomaterials-15-00633-f004:**
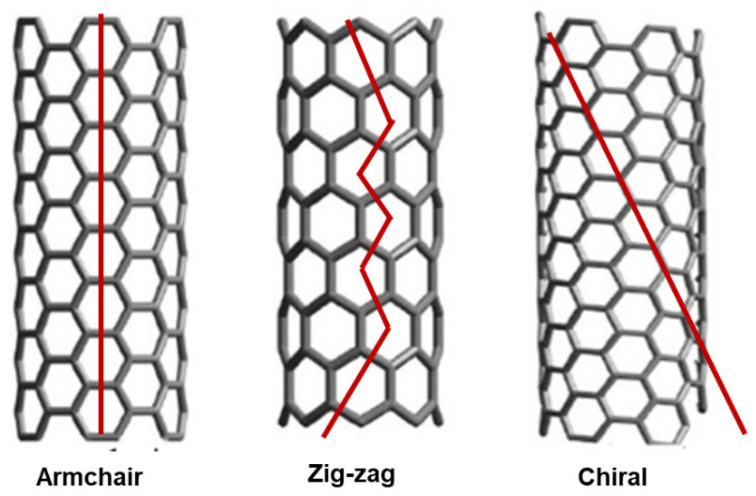
Armchair, zigzag, and chiral structures of SWCNTs. These images by unknown author are licensed under CC BY and are reproduced by our previous work [35].

**Figure 5 nanomaterials-15-00633-f005:**
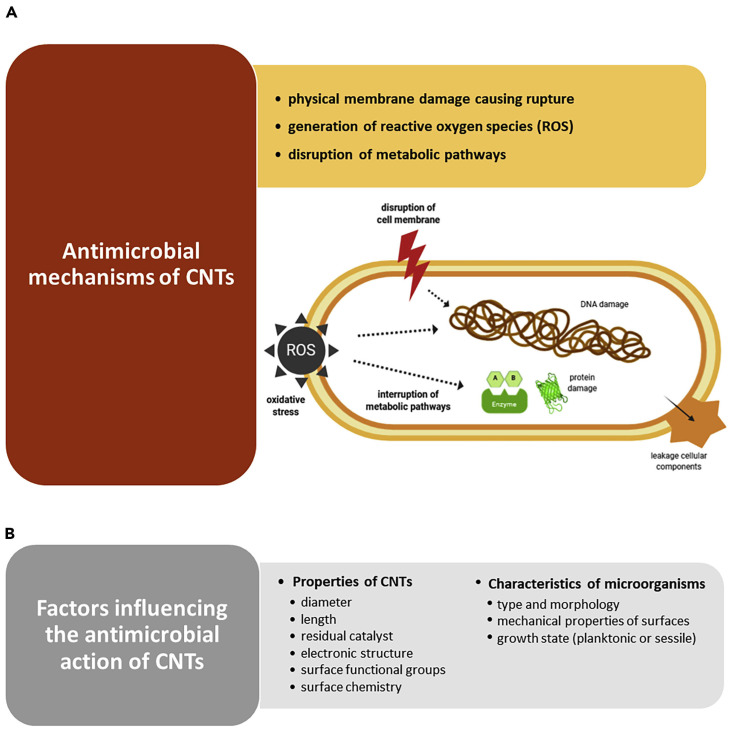
(**A**) Possible mechanisms by which CNTs exert antimicrobial effects: physically piercing of outer membrane, with irreversible membrane damage, cell lysis and release of cytoplasmic content; reactive oxygen species (ROS) overproduction causing oxidative stress (OS), which leads to DNA damage, protein and lipid peroxidation, and impairment of metabolic pathways, which results in cell death. (**B**) Factors influencing the antimicrobial performance of CNTs. This image has been reproduced by an article published in open access [90] and available under the Creative Commons CC-BY-NC-ND (CC BY-NC-ND 4.0, https://creativecommons.org/licenses/by-nc-nd/4.0/, accessed on 18 March 2025) license and permits non-commercial use of the work as published, without adaptation or alteration provided the work is fully attributed.

**Figure 6 nanomaterials-15-00633-f006:**
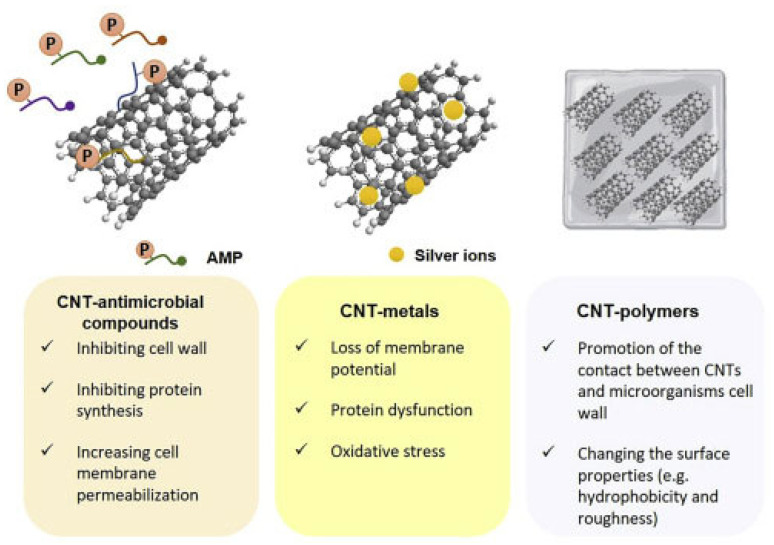
Synergistic effects observed by associating CNTs to AMPs, MNPs, and polymers. This image has been reproduced by an article published in open access [90] and available under the Creative Commons CC-BY-NC-ND (CC BY-NC-ND 4.0, https://creativecommons.org/licenses/by-nc-nd/4.0/, accessed on 18 March 2025) license and permits non-commercial use of the work as published, without adaptation or alteration provided the work is fully attributed.

**Figure 7 nanomaterials-15-00633-f007:**
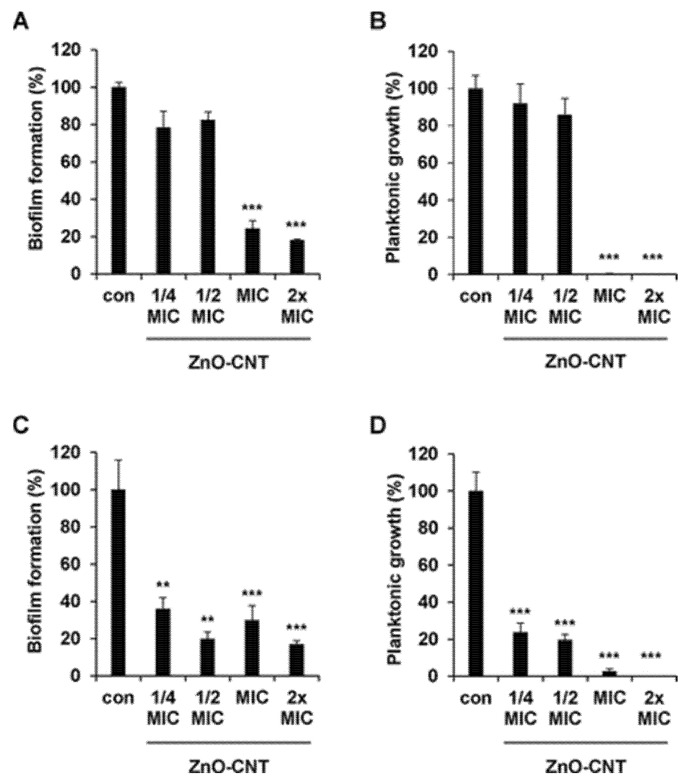
ZnO-CNT nanocomposite with different concentrations, 1/4, 1/2, 1, and 2× MIC, showed biofilm inhibition against (**A**) *E. coli* and (**C**) *P. aeruginosa*. Planktonic growth inhibitions against (**B**) *E. coli* and (**D**) *P. aeruginosa* were determined. Error bars represent the mean ± SD from the average of triplicate experiments. *** *p* < 0.001, ** *p* < 0.01. Reprinted (adapted) with permission from “Copyright 2025 American Chemical Society” [148].

**Table 1 nanomaterials-15-00633-t001:** The antimicrobial performance of pristine carbon nanotubes in different studies.

Types of CNTs	Synthesis Method	Concentration	Species	Main Findings	Ref.
SWCNTs	CO disproportionation	5 µg/mL	*E. coli*	Releasing intracellular content due to irrecoverable outer membrane damage	[26]
SWCNTs	CO disproportionation	5 µg/mL	*E. coli*	Microbial cells lost their cellular integrity	[27]
MWCNTs	CVD method	5 µg/mL	*E. coli*	Many of the bacterial cells remain intact and preserve their outer membrane	[20]
SWCNTs/MWCNTs	CVD method	20, 50, 100 µg/mL	*L. acidophilus*, *E. coli* *B. adolescentis*, *E. faecalis* *S. aureus*	Antimicrobial mechanism associated with length-dependent wrapping and diameter-dependent piercing upon microbial cell membrane damage and the release of intracellular contents	[28]
MWCNTs	Nanocycle productions	1.5–100 mg/L	*E. coli*	Low microbial toxicity	[103]
MWCNTs	-	-	*E. coli*, *B. subtilis* *P. aeruginosa*	2-log cell density reduction in viability of pathogens	[104]
DWCNTs/MWCNTs	NE scientific productions	20/100 µg/mL	*S. aureus*, *P. aeruginosa* *K. pneumoniae*, *C. albicans*	MWNTs’ antimicrobial activity is higher than DWNTs	[97]
MWCNTs	Nanotech Labs productions	20 mg/20 mL	*P. fluorescens*	44% of inactivated bacteria MWNTs showed a significant effect on the inhibition of microbial adhesion due to the electrochemical potential	[105]
SWCNTs	-	5 µg/mL	*E. coli*, *B. subtilis*	No physical destruction was observed below 10 nN of applied force	[94]
SWCNT/DWCNT MWCNT	Electric arc discharge and CCVD	100 µg/mL	*S. aureus*, *P. aeruginosa*, *C. albicans*	Microbial death induced by the aggregation of CNTs trapped on the microbial cell surface	[93]
SWCNTs/MWCNTs	-	0.2 mg/mL	*E. coli*	Control bacteria grow by laser-activated CNTs	[106]
SWCNT-OHs	-	50 to 250 µg/mL	*Salmonella*	−7log reduction in cell viability at 200–250 µg/mL	[95]

CO = Carbon oxide; CVD = chemical vapor deposition; CCVD = catalytic chemical vapor deposition.

**Table 2 nanomaterials-15-00633-t002:** Overview of the antimicrobial activity of functionalized SWCNT-based nanocomposites in different studies.

Material Blend	Concentration	Species	Main Findings	Ref.
SWCNTs-OH, -COOH, -NH_2_	50–200 µg/mL	*S. aureus*, *B. Subtilis*, *S. typhimurium*	SWCNTs-OH and -COOH showed higher microbial inhibition rates (7-log reduction) than SWCNTs-NH_2_	[95]
Ag-SWCNTs containing TP226, TP359, TP557 peptides	5 µg/mL	*S. aureus*	In skin models treated with silver-SWCNTs antimicrobial activity of only 1-log reduction was observed	[114]
SWCNTs functionalized with DNA and LSZ	~25 mg/L	*S. aureus*, *M. lysodeikticus*	DNA- and LSZ-SWCNTs caused 84% microbial reduction	[127]
SWCNTs-PLGA complexes	<2% by weight	*E. coli*, *S. epidermidis*	SWCNTs-PLGA caused a 98% reduction in metabolic activity	[92]
SWCNTs- PVK nanocomposite	3 wt.%	*E. coli*, *B. subtilis*	SWCNTs-PVK induced 90% and 94% of *B. subtilis* and *E. coli* inhibition in the planktonic cells and showed a significant reduction of biofilm formation	[130]
SWCNTs-PGA/PLL (layer-by-layer)	<2% by weight	*E. coli*, *S. epidermidis*	SWCNTs/PGA/PLL showed a 90% reduction in pathogens	[92]
Oxidized SWCNTs-PVOH) nanocomposite	0–10% (*w*/*w*)	*P. aeruginosa*	O-SWCNTs-PVOH gradually decreased viability of cells with an increase in nanotube loading	[132]
SWCNTs/porphyrin composite	0.04 mg/mL	*S. aureus*	SWCNTs/porphyrin caused visible light-induced damage to the cell membrane	[133]
SWCNTs-PEG)/poly-(ε caprolactone) composites	0.5–1.0 wt.%	*P. aeruginosa*, *S. aureus*	SWCNT/copolymer complex caused lower bacteria inhibition than pure polymer complex	[135]
SWCNTs-polyamide membranes	0.1–0.2 mg/mL	*E. coli*	Nanocomposite inactivated the microbial cells by 60% after 1 h of contact time	[133]

LSZ = lysozyme; PLGA = poly-(lactic-co-glycolic acid); PVK = polyvinyl-N-carbazole; PGA = poly-(L-glutamic acid); PLL = poly-(L-lysin); PVOH = poly-(vinyl alcohol); PEG = poly-(ethylene glycol).

**Table 3 nanomaterials-15-00633-t003:** Overview of the antimicrobial activity of functionalized MWCNT-based nanocomposites in different studies.

Material Blend	Concentration	Species	Main Findings	Ref.
MWCNTs-OH, -COOH, -NH_2_	50–200 µg/mL	*S. aureus*, *B. subtilis* *S. typhimurium*	MWCNTs-OH and -COOH did not significantly induce antimicrobial activity	[95]
	25 µg/mL	*E. coli*, *B. subtilis*, *S. aureus*	MWCNTs-COOH caused 30, 40, 50% inactivation for *B. subtilis*, *E. coli*, *S. aureus*, respectively	[136]
	20 µg/mL	*S. aureus*, *E. coli* *P. aeruginosa*	MWCNTs-COOH caused 27, 34, 23% inactivation for *P. aeruginosa*, *E. coli*, *S. aureus*	[137]
	20 mg/20 mL		MWCNTs-COOH inactivated the bacterial cells by 27, 20, 15% for *P. aeruginosa*, *E. coli*, *S. aureus*	[138]
	20, 50, 100 µg/mL	*E. coli*, *S. aureus*, *E. faecalis* *L. acidophilus*, *B. adolescentis*	MWCNTs-COOH and MWNTs-OH induced dose-dependent microbial inhibition	[97]
	1000 µg/mL	*V. parahaemolyticus*	Time-dependent antimicrobial activity CNTs did not pierce the cell membrane CNTs wrapped around the surface of pathogens	[153]
Surfactant-functionalized MWCNTs with SDBS, SC, SDS TX-100, DTAB, CTAB, PVP	0–100 mg/mL	Group A *Streptococcus*	Carboxylate-MWCNTs with antibodies mitigate soft tissue infections	[154]
	20–100 µg/mL	*S. aureus*, *P. aeruginosa* *K. pneumoniae*, *C. albicans*	Non-covalently dispersed CNTs inhibited bacteria by a time and concentration-dependent mechanism	[97]
	1.0, 0.5, 0.25, 0.125 mg/mL	*S. mutans*	Functionalized MWCNTs caused cell membrane rupture via direct contact	[155]
	0.1, 0.5, 1 mg/mL	*E. coli*	Functionalized MWCNTs penetrated the bacterial cell membrane due to electrostatic forces	[156]
AgNP-coated MWCNTs	2–30 wt.%		Cell membrane of bacteria damaged via direct contact	[157]
ZnO-MWCNTs	0.0078, 1 wt.% 0.00625, 0.0039 wt.%	*E. coli*, *P. aeruginosa* *E. faecalis*, *S. aureus*	ZnO-CNT urinary catheter inhibited biofilm formation by 53.42% and 56.44% after 120 h against *E. coli* and *P. aeruginosa*	[148]
MWCNTs-lysine	0.01875–0.6 mg/mL	*S. aureus*, *E. coli*, *S. agalactiae* *S. typhimurium*, *S. dysgalactiae* *K. pneumoniae*	Electrostatic adsorption	[158]
MWCNTs-PPI	25 µg/mL	*E. coli*, *B. subtilis*, *S. aureus*	MWCNT nanocomposite caused 97, 87% inactivation for *S. aureus/B. subtilis*, and *E. coli*	[136]
MWCNTs-aromatic dendrimer polyamide	20 μg/mL	*S. aureus*, *E. coli*, *P. aeruginosa*	MWCNT nanocomposite caused 36, 65, 73% inactivation for *S. aureus*, *P. aeruginosa*, *E. coli*	[137]
PAMAM-grafted MWCNTs	20 mg/20 mL	*S. aureus*, *E. coli*, *P. aeruginosa*	MWCNT nanocomposite caused 60, 34, 23% for *P. aeruginosa*, *E. coli*, *S. aureus*	[138]
Oxidized MWCNTs/PVOH nanocomposite	0–10% (*w*/*w*)	*P. aeruginosa*	MWCNTs–PVOH reduced bacteria viability by increasing concentrations	[132]
MWCNTs–chitosan hydrogels	25, 50, 100 mg/40 mL	*S. aureus*, *E. coli*, *C. tropicalis*	MWCNTs–chitosan hydrogels exhibited higher antimicrobial activity against *S. aureus* and *C. tropicalis* than *E. coli*	[151]
	0.01%, 0.1% and 0.2% (*w*/*w*)	*E. coli*, *S. pneumoniae* *S. racemosum*, *C. albicans P. aeruginosa* *G. candidum*, *A. fumigatus*	MWCNT nanocomposite showed a higher microbial inhibition rate against Gram-positive bacteria	[152]

PPI = poly-(propyleneimine); PAMAM = poly(amidoamine); PVOH = poly-(vinyl alcohol); SDBS = sodium dodecylbenzene sulfonate; SC = sodium cholate; SDS = sodium dodecyl sulphate; TX-100 = triton X-100 (TX-100); DTAB = dodecyltrimethylammonium bromide; CTAB = cetyltrimethylammonium bromide; PVP = polyvinylpyrrolidone.

**Table 4 nanomaterials-15-00633-t004:** In vitro cytotoxicity of CNTs for early experiment of years 2003–2011.

Nanomaterials	Cell Lines	Observation	Ref.
Metal oxide nanoparticles and SWCNTs	A549	Penetrate the cell	[167]
CNTs	Rat macrophages (NR8383), A549	Increase in intracellular reactive oxygen species	[168]
CNTs	MSTO-211H	Agglomerated CNTs show greater cytotoxicity	[169]
MWCNTs	HEK	Decrease in cell viability and increase in IL-8	[170]
Functionalized SWCNTs	Human fibroblasts 3T6 and murine 3T3	Pass through the cellular membrane and concentrate in the cytoplasm	[171]
SWCNTs	HaCaT, HeLa, H1299, and A549	Increase in oxidative stress and inhibition of cell proliferation	[172]
SWCNTs	HaCaT	Cell death, oxidative stress, and increase in lipid peroxides	[173]
SWCNTs	HaCaT and BEAS-2B	Loss of cellular integrity and cellular apoptosis	[184]
SWCNTs	Lymphocytes and macrophages	Uptake of SWCNTs	[175]
MWCNTs	HEK	Cell-cycle inhibition	[188]
Functionalized SWCNTs	Human dermal fibroblasts	Less toxicity	[176]
SWCNTs-streptavidin complex	HL60	Low toxicity	[177]
CNTs	Lymphocytes	Increase the secretion of cytokines	[179]
SWCNTs	HEK293T	Inhibition of cell proliferation and decrease in cell adhesive ability	[180]
SWCNTs and MWCNTs	Alveolar macrophages	SWCNTs showed higher toxicity than MWCNTs	[181]
Purified SWCNTs	Lung fibroblast V79	DNA damage	[183]
SWCNTs	RAW 264.7	Production of TGF-β1	[174]
Iron-rich SWCNTs	Macrophages	Phagocytosis of the SWCNTs and conversion of extracellular O^2−^ into hydroxyl radicals	[184]
Ground MWCNTs	Rat peritoneal macrophages	Cytotoxicity and overproduction of proinflammatory cytokines	[185]
MWCNTs	J774.1	Cytotoxic effects by the rupturing of plasma membrane	[186]
CNTs	Human aortic endothelial cells	Increase in the MCP-1, VCAM-1, and IL-8	[184]
MWCNTs	Alveolar macrophages	Cell death	[192]
Pristine MWCNTs and oxidized MWCNTs	T lymphocytes	Apoptosis	[193]

CNT = carbon nanotube; HEK = human epidermal keratinocyte; MWCNT = multi-walled carbon nanotube; SWCNT = single-walled carbon nanotube.

**Table 5 nanomaterials-15-00633-t005:** In vivo case studies on pulmonary toxicity of CNTs.

Nanomaterials	Animals	Observation	Reference
SWCNTs	Mice	Peri-bronchial inflammation and necrosis	[194]
Pristine SWCNTs	Rat	Inflammation and multifocal granulomas	[195]
SWCNTs/SiO_2_	Mice	Granulomas and lung fibrosis	[174]
MWCNTs	Rat	Pulmonary fibrosis	[185]
SWCNTs	Mice	Inflammatory response, OS, collagen deposition	[196]
MWCNTs	Mice	Inflammation and granulomas	[32]
MWCNTs	Rat	Granuloma and collagen depositions	[203]
MWCNTs/SWCNTs	Guinea pigs	Pneumonitis	[204]
MWCNTs	Rat	Inflammation, granuloma, and lung fibrosis	[205]

CNT = carbon nanotube; MWCNT = multi-walled carbon nanotube; SWCNT = single-walled carbon nanotube; OS = oxidative stress.

**Table 6 nanomaterials-15-00633-t006:** In vitro and in vivo studies for assessing the possible cytotoxicity and genotoxicity by exposure to CNTs.

Theme of the Study	Tested Cells	Toxic Effects	Findings	Refs.
Genotoxicity of MWCNTs in human cells	HeLa, MCF7 Human respiratory epithelial cell	Genotoxicity	DNA damage, chromosomal aberrations, OS by exposure to CNTs	[210,211]
Inhalation exposure to CNTs induces pulmonary toxicity	Mice	Pulmonary toxicity	Inflammation, granuloma formation, lung fibrosis upon inhalation ⬆ Levels of cytokines	[194,212]
Skin exposure to CNTs to assess dermal toxicity	Murine epidermal cells (JB6 P+), and immune-competent hairless SKH-1 mice	Dermal toxicity	Direct skin exposure to CNTs led to irritation and inflammation Penetration into deeper skin layers is still debated	[213]
Impact of CNTs on immune system	Lymphocytes, T cells, monocytes, and dendritic cells	Immunotoxicity	CNTs affected immune responses Altered cytokine production and immune cell functions	[214,215]
Cardiovascular effects of CNTs in animal models	Mice	Cardiovascular toxicity	Inflammation- and OS-induced increased blood pressure and cardiovascular diseases	[216]
Liver toxicity/biodistribution of CNTs in mice	Mice	Hepatoxicity	Liver damage and alterations in liver enzyme levels	[217]
Renal toxicity of MWCNTs in rats	Human embryonic kidney (HEK293) cells	Renal toxicity	CNT accumulation in the kidneys, renal inflammation, OS Kidney functional impairment	[218]
OS induced by CNTs	Alveolar macrophage (AM)	OS	CNTs generate ROS, OS, cell and tissue damage	[181]
In vitro neurotoxicity of CNTs	Mammalian cell lines	Neurotoxicity	Neuroinflammation and neurons damage	[219]

OS = Oxidative stress; ROS = reactive oxygen species; ⬆ indicated increased.

**Table 7 nanomaterials-15-00633-t007:** A quick overview of CNTs’ toxicity on various organs by in vivo and in vitro studies.

CNTs	Tested Cells	Affected Organ	Results	Ref.
MWCNTs	Male Sprague Dawley rats	Nervous system	Dramatic alterations of sympathetic and parasympathetic nervous system’s equilibrium	[220]
MWCNTs	Mice	Nervous System and BBB	Acute lung exposure to MWCNTs damaged BBB integrity, induced nerve inflammatory responses	[221]
CNTs	Male NMRI mice	Nervous system	Behavioral toxicity with manifestation of sadness or anxiety	[222]
SWCNTs	PC-12 cells	Nervous system	Toxicity to PC-12 cells ⬆ Harmful to differentiated PC-12 cells	[223]
SWCNTs	Male C57BL/6 mice	Pulmonary immune system	CNTs make people more vulnerable to respiratory virus infections	[224]
SWCNTs	Six-week-old specific-pathogen-free ICR mice	Immune system Reproductive system	CNTs affected development and reproduction and produced immunological toxicity	[225]
SWCNTs	BALB/c macrophage cells J774A and BALB/c mice	Immune system	Immune toxicity *	[226]
MWCNTs	T lymphocytes	Immune system	Concentration-dependent harmful of CNTs to human T cells	[193]
SWCNTs	6/8 weeks old females of the CD1 outbred strain Mouse ES cell line D3 NIH3T3 cells	Embryos	SWCNTs can cause harm to mammalian embryos	[207]
CNTs	Kunming mice	Embryos	CNTs harmed fetuses, cause miscarriage, and were harmful to embryos	[227]
MWCNTs	Zebrafish embryo	Embryos	Significant length-dependent development risk	[228]
CNTs	Mouse embryonic fibroblasts (MEFs) and C57BL/6J mice	Embryos	Induction of hereditary embryotoxicity	[229]
O-SWCNTs	Artemia salina	Embryos	Large amount generation of ROS and deformed Artemia salina	[230]
CNTs	Male BALB/c mice	Genitals	CNTs poison mice’s reproductive organs	[208]
SWCNTs/MWCNTs	MeT-5A and BEAS 2B cells	Genome	DNA damage in MeT-5A cells by both MWCNTs and SWCNTs	[231]

ROS = reactive oxygen species; ⬆ indicated increased, high, higher; * there was a negative correlation between immune toxicity and SWCNT dispersion; O-SWCNTs = Oxidized SWCNTs.

**Table 8 nanomaterials-15-00633-t008:** Strategies to reduce CNT toxicity.

Strategy	Goal	Modifying Agents/Methods	Results	Refs.
Modify CNTs surface with biocompatible materials or othermolecules	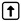 Dispersion in biological fluidsInfluenced cellular uptake 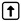 Solubility 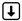 Toxicity	Proteins, surfactants	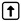 Tumor targeting 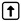 Therapeutic benefits 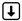 Toxicity	[238,239,240]
Folic acid	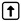 In vivo tumor targeting 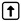 Therapeutic benefits 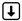 Toxicity	[241]
Polyacrylamide hydrogels *Biomaterial, TiO_2_	100% survival of L929 mouse fibroblast	[242]
Application of coatings to CNTs	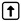 CNTs biocompatibility 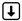 Potential toxicityPrevent direct contact withbiological systems 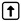 CNTs solubility	Curcumin lysine **	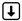 IL-6, IL-8, IL-1β, TNFα, N-FκB 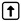 Antioxidant enzyme catalase 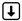 ROS generationRecovery of mitochondrial membranepotential 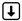 Cell death	[243]
Encapsulation of CNTsUsing CNTs to entrap bioactive molecules	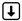 Direct cells exposure to CNTs Control of CNTs release 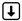 CNTs impact on tissues	PEG (entrapping agent)Oxaliplatin (entrapped agent)	PEGylation delayed oxaliplatin release rate 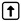 Drug’s anticancer effects on HT-29 cells	[244]
Tailor the diameter size and length of CNTs	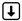 Toxicological impact	N.A.	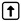 Specific surface area 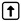 Transmembrane mobility 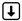 Toxicity	[245]
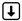 Harm to lysosomes for large-diameters MWCNTs	[246]
Optimization ofpurification processes	Remove metal impuritiesRemove residual catalysts	Chemical/electrochemical oxidation [247]High chlorine partial pressure [248]Microwave-assisted digestion [249]Incandescent annealing	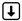 Lower harmful effects	[250]
Engineering controls Suitable PPE	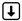 Inhalation exposure	Proper ventilation/respiratory protection	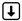 Risk of respiratory toxicity	N.R.
Co-administration of CNTs with antioxidants	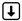 Potential OS 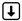 Cellular damage	Quercetin	Prevention of the oxidative damage 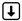 Inflammatory effects 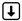 Immuno-toxic effects	[251]

* Encapsulation agent for CNTs-COOH; ** used to coat MWCNTs; N.A. = not applicable; N.R. = not reported; 
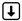
 indicates minor reduction, lower, decreased, decrease; 
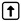
 indicates improved, increase, increased, major; PPE = personal protective equipment; OS = oxidative stress.

**Table 9 nanomaterials-15-00633-t009:** Published and under development * ISO standards concerning CNTs and CNT-containing nanocomposites.

Standard	Description
ISO/TS 19808:2020	Specifies the characteristics of suspensions containing MWCNTs to measure and the corresponding measure methods
ISO/TS 23690:2023	Specifies a mild oxidation method by TGA to determine the content of carbon impurities (amorphous carbon, other types of structured carbon)
ISO/TS 11308:2020	Guidelines for the characterization of CNT-containing samples and impurities by TGA in inert or oxidizing environment
ISO/TR 23463:2022	Reviews correct characterization of CNT and CNF aerosols for inhalation Establishes characterization required to assess their inhalation toxicity
ISO/TS 11888:2017	Describes methods for characterizing mesoscopic shape factors of MWCNTs by SEM, TEM, viscometry, DLS
ISO/TR 10929:2012	Identifies the basic properties of MWCNTs and the content of impurities Highlights the available measurement methods to industry for their determination
ISO/TS 10797:2012	Establishes methods for characterizing the morphology of SWCNTs, identifying the elemental composition of other materials in SWCNT samples by TEM and EDXRD
ISO/TS 10867:2019	Gives guidelines for the determination of the chiral indices of the semi-conducting SWCNTs by NIR-PL
ISO/TS 10868:2017	Provides methods to characterize the diameter, purity, and other characteristics of compounds containing SWCNTs by OAS
ISO/TS 10798:2011	Establishes methods to characterize morphology and identify the elemental composition of catalysts and other inorganic impurities in RSWCNTs and PSWCNT powders and films by SEM, EDXRD
ISO/TS 13278:2017	Provides methods for the determination of residual elements other than carbon in samples of SWCNTs and MWCNTs by ICPMS
ISO/TS 11251:2019	Specifies method for the characterization of evolved gas components in SWCNT samples by EGA/GCMS
ISO 20523:2017	Specifies classification, designations, and short names for CNT-based films deposited by PVD or CVD
ISO/AWI 11308 *	Gives guidelines for the characterization of CNT-containing samples by TGA in inert or oxidizing environment Provides guidelines on the purity assessment of the CNT samples through quantitative measures of the carbon and non-carbon species present
ISO/AWI TS 21497 *	Describes a method to remove CNTs, NG, and CNHs from industrial and research wastewater by using hypochlorite compounds

CNHs = Carbon nano-horns; NG = nano graphene; TGA = thermogravimetric analysis; PVD = physical vapor deposition; CVD = chemical vapor deposition; EGA/GCMS = evolved gas analysis/gas chromatograph mass spectrometry; ICPMS = inductively coupled plasma mass spectroscopy; RSWCNTs = raw SWCNTs; OAS = optical absorption spectroscopy; NIR-PL = near infrared photoluminescence spectroscopy; EDXRD = energy dispersive X-ray diffraction; SEM = scanning electron microscopy; TEM = transmission electron microscopy; DLS = dynamic light scattering analysis; * under development (all other standards are published).

## Data Availability

Not applicable.

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
