# Peer review of "Antimicrobial Nanotubes: From Synthesis and Promising Antimicrobial Upshots to Unanticipated Toxicities, Strategies to Limit Them, and Regulatory Issues"

_nanomaterials, 2025, doi:10.3390/nano15080633_

Round 1

Reviewer 1 Report

Comments and Suggestions for Authors

The review presents Synthesis, Biomedical Applications, Toxicity Issues of CNTs, Possible Strategies to Moderate CNTs’ Toxic Effects, Regulatory Considerations. There are some issues as the following.

  1. More figures should be presented in the review to demonstrate the important result and findings in the relevant references.
  2. Authors should highlight what makes the work stand out in relation to other review articles already published.
  3. The manuscriptpresents long texts about what some references present, but fails to compile and them. A more robust discussion in these areas would have contributed to a more comprehensive understanding of antimicrobial nanotubes.
  4. The review mainly presents Synthesis, Biomedical Applications, Toxicity Issues of CNTs, Possible Strategies to Moderate CNTs’Toxic Effects, Regulatory Considerations. The title of the manuscript should revised to comply with the text.
Comments on the Quality of English Language

 The English could be improved to more clearly express the research.

Author Response

The review presents Synthesis, Biomedical Applications, Toxicity Issues of CNTs, Possible Strategies to Moderate CNTs’ Toxic Effects, Regulatory Considerations. There are some issues as the following.

  1. More figures should be presented in the review to demonstrate the important result and findings in the relevant references.

We appreciate the relevant suggestion by the Reviewer. Her/his request has been satisfied by inserting in the revised Review, three additional Figures (Figure 5, 6 and 7) with related references and copyright permissions. Such new Figures have been included in the Section concerning the CNTs antimicrobial activities, which is the main topic of this paper. Please, consider lines 320-321, 323-331 and 373-392 for Figure 5, lines 415-417 and 419-423 for Figure 6, and 558-563, 565-571 for Figure 7.

  1. Authors should highlight what makes the work stand out in relation to other review articles already published.

We thank the Reviewer for her/his suggestion. In this regard, we have first extended the Introduction section providing a background also to the potential toxicity of CNTs, the developed strategies to limit it, and regulatory issues, which was missing. Additionally, a brief description of review’s contents, its relevance and overall merits have been included in the last part of Introduction. Please, consider lines 115-165.

  1. The manuscript presents long texts about what some references present, but fails to compile and them. A more robust discussion in these areas would have contributed to a more comprehensive understanding of antimicrobial nanotubes.

We thank the Reviewer for her/his suggestion again. In this regard, we have extended the discussion on the antimicrobial properties demonstrated by CNTs in the studies reported in this review (Table 1, 2 and 3), providing additional summarizing comments to the results collected in Tables, thus also evidencing pro and cons deriving by CNTs modifications. Please, see lines 398-409 and 608-632.

  1. The review mainly presents Synthesis, Biomedical Applications, Toxicity Issues of CNTs, Possible Strategies to Moderate CNTs’Toxic Effects, Regulatory Considerations. The title of the manuscript should revised to comply with the text.

We thank a lot the Reviewer for the suggestion, which gave us the possibility to make the Title of this paper more compliant with its content, thus better attracting the readers’ attention. The title has been revised (lines 2-4).

Comments on the Quality of English Language

The English could be improved to more clearly express the research.

Upon suggestion of the Reviewer, all manuscript has been revised by our colleague prof. Deirdre Kantz, English mother tongue teacher, working for University of Genoa and Pavia, to reduce grammar errors and typos.

Reviewer 2 Report

Comments and Suggestions for Authors

This review article by Silvana Alfei et al. systematically summarized carbon nanomaterials, especially carbon nanotubes (CNTs) including single-walled CNTs (SWCNTs) and multi-walled CNTs (MWCNTs), with their preparation strategies, biomedical applications, toxicity issues, strategies against CNTs’ toxic effects, and regulatory considerations. This review affords systematic and well-organized information on CNTs and their potential applications. I recommend its publication in Nanomaterials. The authors are suggested to consider the following comments to further improve the quality of this manuscript.

  1. Check again the Section 5.1, and section 6.1, the authors do not need to include these two subtitles, please merge the contents in section 5.1 to section 5, and the contents in section 6.1 to section 6.

  2. As for the perspectives, the authors should mention the wide applications of CNTs in other fields, such as CO2/CO electroreduction, water splitting, N2 reduction, and etc. And give a brief discussion on this part.

  3. For the review title, “Antimicrobial Nanotubes Between Promising Outcomes, Unanticipated Toxicities, Strategies to Limit Them and Regulatory Issues: A Review”, the authors use between, do they mean among or for? Please give more clear and easy understanding information in the title.

Author Response

This review article by Silvana Alfei et al. systematically summarized carbon nanomaterials, especially carbon nanotubes (CNTs) including single-walled CNTs (SWCNTs) and multi-walled CNTs (MWCNTs), with their preparation strategies, biomedical applications, toxicity issues, strategies against CNTs’ toxic effects, and regulatory considerations. This review affords systematic and well-organized information on CNTs and their potential applications. I recommend its publication in Nanomaterials. The authors are suggested to consider the following comments to further improve the quality of this manuscript.

These authors are very grateful to the Reviewer for the positive comments and for the following suggestions, which we hope to have correctly and exhaustively satisfied.

Check again the Section 5.1, and section 6.1, the authors do not need to include these two subtitles, please merge the contents in section 5.1 to section 5, and the contents in section 6.1 to section 6.

Done.

As for the perspectives, the authors should mention the wide applications of CNTs in other fields, such as CO2/CO electroreduction, water splitting, N2 reduction, and etc. And give a brief discussion on this part.

We are very grateful to Reviewer for her/his help in ameliorating our work, with this suggestion. A brief discussion on topic suggested by the Reviewer has now included in the Conclusions Section (lines 1267-1274).

For the review title, “Antimicrobial Nanotubes Between Promising Outcomes, Unanticipated Toxicities, Strategies to Limit Them and Regulatory Issues: A Review”, the authors use between, do they mean among or for? Please give more clear and easy understanding information in the title.

We are grateful to Reviewer for her/his help in ameliorating the Title of our work, thus making it clearer and more in compliance with the Review contents (lines 2-4).

Reviewer 3 Report

Comments and Suggestions for Authors

The manuscript, entitled "Antimicrobial Nanotubes Between Promising Outcomes, Unanticipated Toxicities, Strategies to Limit Them and Regulatory Issues: A Review" is a thorough and contemporary review. It thoroughly explores the synthesis, antimicrobial properties, functionalization strategies, toxicological concerns, and regulatory issues surrounding carbon nanotubes (CNTs). The review is supported by a substantial body of research and case-specific tables that enhance its readability and clarity. However, a few areas could be improved to further strengthen the manuscript.

  • First, the review would benefit from a more concise presentation, particularly in the sections addressing synthesis and antimicrobial mechanisms, where some information is repetitive. Streamlining these parts would improve clarity and maintain reader engagement.
  • Second, the section on regulatory issues is currently underdeveloped. Expanding this part by referencing key international standards—such as REACH, FDA, and ISO—and identifying existing regulatory gaps would provide a more robust and informative discussion of the broader context surrounding CNT applications.
  • Third, to present a more balanced and critical perspective, the authors are encouraged to discuss conflicting toxicological findings and elaborate on the practical challenges of scaling up CNT-based technologies. Including these aspects would provide a more nuanced view and strengthen the review’s contribution to the field.
  • The manuscript would benefit from a language revision. Certain expressions are awkward or overly complex. Simplifying lengthy sentences and replacing terms like "knowhows" with "technologies" would enhance overall readability.

Author Response

The manuscript, entitled "Antimicrobial Nanotubes Between Promising Outcomes, Unanticipated Toxicities, Strategies to Limit Them and Regulatory Issues: A Review" is a thorough and contemporary review. It thoroughly explores the synthesis, antimicrobial properties, functionalization strategies, toxicological concerns, and regulatory issues surrounding carbon nanotubes (CNTs). The review is supported by a substantial body of research and case-specific tables that enhance its readability and clarity. However, a few areas could be improved to further strengthen the manuscript.

  • First, the review would benefit from a more concise presentation, particularly in the sections addressing synthesis and antimicrobial mechanisms, where some information is repetitive. Streamlining these parts would improve clarity and maintain reader engagement.

We greatly appreciated the Reviewer suggestions, which have been considered. Concerning the section addressing the synthesis of CNTs, it has been streamlined and shortened, directing readers interested in further details to consider Table A2 (reported in the Appendix A in this work) and our previous paper (Ref. 35). Please, consider lines 190-215. Concerning, the antimicrobial mechanisms, to satisfy both the requests of this Reviewer and those of another one, requiring more Figures, the clarity and readability of this part has been streamlined by adding new eyes-catching Figures 5 and 6. Repetitions were instead removed.

  • Second, the section on regulatory issues is currently underdeveloped. Expanding this part by referencing key international standards—such as REACH, FDA, and ISO—and identifying existing regulatory gaps would provide a more robust and informative discussion of the broader context surrounding CNT applications.

On suggestion of the Reviewer, who we thank, the section on regulatory issue has been modified and extensively expanded by reporting different International Organizations developing key international standards for materials production and characterization and by considering specifically ISO and ASTM and related standards developed concerning nanomaterials and CNTs. Please, see lines 1123-1205 and new Table 9.

  • Third, to present a more balanced and critical perspective, the authors are encouraged to discuss conflicting toxicological findings and elaborate on the practical challenges of scaling up CNT-based technologies. Including these aspects would provide a more nuanced view and strengthen the review’s contribution to the field.

We thank the Reviewer for her/his suggestions which we have appreciated and addressed. The additional discussions suggested by the Reviewers have been added to the Section 7, retitled “Conclusions and Perspectives. Please, see lines 1248-1255 and 1256-1264.

  • The manuscript would benefit from a language revision. Certain expressions are awkward or overly complex. Simplifying lengthy sentences and replacing terms like "knowhows" with "technologies" would enhance overall readability.

Upon suggestion of the Reviewer, all manuscript has been revised by our colleague prof. Deirdre Kantz, English mother tongue teacher, working for University of Genoa and Pavia, to reduce grammar errors and typos. As asked, the term “knowhows” has been replaced with “technologies” (line 38).

Round 2

Reviewer 1 Report

Comments and Suggestions for Authors

The problems in the manuscript have been solved and the reviewer think that it should be accepted.